# Crosstalk with keratinocytes causes GNAQ oncogene specificity in melanoma

Oscar Urtatiz[1], Amanda Haage[2], Guy Tanentzapf[2], Catherine D Van Raamsdonk[1]*

[1]Department of Medical Genetics, Life Sciences Institute, University of British Columbia, Vancouver, Canada; [2]Department of Cellular and Physiological Sciences, Life Sciences Institute, University of British Columbia, Vancouver, Canada

**Abstract** Different melanoma subtypes exhibit specific and non-overlapping sets of oncogene and tumor suppressor mutations, despite a common cell of origin in melanocytes. For example, activation of the $G\alpha_{q/11}$ signaling pathway is a characteristic initiating event in primary melanomas that arise in the dermis, uveal tract, or central nervous system. It is rare in melanomas arising in the epidermis. The mechanism for this specificity is unknown. Here, we present evidence that in the mouse, crosstalk with the epidermal microenvironment actively impairs the survival of melanocytes expressing the GNAQ$^{Q209L}$ oncogene. We found that GNAQ$^{Q209L}$, in combination with signaling from the interfollicular epidermis (IFE), stimulates dendrite extension, leads to actin cytoskeleton disorganization, inhibits proliferation, and promotes apoptosis in melanocytes. The effect was reversible and paracrine. In contrast, the epidermal environment increased the survival of wildtype and Braf$^{V600E}$ expressing melanocytes. Hence, our studies reveal the flip side of $G\alpha_{q/11}$ signaling, which was hitherto unsuspected. In the future, the identification of the epidermal signals that restrain the GNAQ$^{Q209L}$ oncogene could suggest novel therapies for *GNAQ* and *GNA11* mutant melanomas.

## Editor's evaluation

Urtatiz and colleagues propose that gain-of-function mutations affecting the G-α-q signaling pathway are not tolerated in melanocytes residing in the interfollicular epidermis because of paracrine signals from neighboring keratinocytes. This is an interesting and important hypothesis that would explain a mystery to the melanoma field.

*For correspondence:
cvr@mail.ubc.ca

Competing interest: The authors declare that no competing interests exist.

## Introduction

*GNAQ* and *GNA11* encode heterotrimeric G protein alpha subunits ($G\alpha_q$ and $G\alpha_{11}$) that are best known for signaling through phospholipase C-beta (PLC-β) to release intracellular calcium stores (***Runnels and Scarlata, 1999***). The somatic substitution of glutamine 209 or arginine 183 in either *GNAQ* or *GNA11* generates constitutive activity of the G protein and is a characteristic early event in uveal melanoma, occurring in up to 90% of cases (***Van Raamsdonk et al., 2010***). A subset of cases without *GNAQ* or *GNA11* mutations exhibits a gain-of-function hotspot mutation in *PLCB4* (***Johansson et al., 2016***; ***van de Nes et al., 2017***; ***Phan et al., 2021***). These mutations are also frequent in blue nevi and primary melanomas of the central nervous system (***Van Raamsdonk et al., 2010***; ***Van Raamsdonk et al., 2009***; ***Küsters-Vandevelde et al., 2010***). Activation of the pathway in uveal melanoma drives cell proliferation and stimulates the mitogen activated protein kinase ('MAPK') pathway through RasGRP3 (***Chen et al., 2017***). $G\alpha_{q/11}$ activation also activates the Hippo pathway through nuclear localization of YAP1 via a Trio-Rho/Rac signaling circuit (***Feng et al., 2014***). On the other hand, Q209 and R183 mutations in either *GNAQ* or *GNA11* are rare in melanocytic neoplasms arising in the epithelium, which are more likely to carry mutations in the MAPK pathway components,

*BRAF*, *NRAS,* and *NF1* (*Van Raamsdonk et al., 2010*; *Fecher et al., 2008*; *Nissan et al., 2014*). The underlying mechanism for the restriction of *GNAQ* and *GNA11* mutations to non-epithelial melanomas is unknown.

There are several possible, non-exclusive explanations for this phenomenon. First, an over-representation of mutations in specific genes in certain neoplasms could be related to exposure to different mutagens in different parts of the body. Concerning melanoma, the skin areas exposed to UV radiation via sunlight accumulate many somatic mutations; characteristically, these are C to T transitions (*Pfeifer et al., 2005*). Most UV radiation does not penetrate very deeply into the body and some areas of the body are rarely exposed, which could generate a varying spectrum of mutations (*Hodis et al., 2012*).

Second, melanocytes arise from neural crest cells all along the anterior-posterior axis. The molecular pathways for melanocyte cell fate determination along this axis are poorly understood (*Soldatov et al., 2019*). However, there are demonstrated differences in the potential for melanomagenesis of melanocytes arising at different anterior-posterior positions in the mouse embryo (*Urtatiz et al., 2020*). It is possible that the developmental origin of a melanocyte could influence its migration during embryogenesis, its differentiation potential, and program long-term gene expression patterns. Therefore, some melanocytes might only respond to specific driver mutations due to intrinsic differences generated during development.

Last, different microenvironments surround melanocytes located in different parts of the body, which could impact the effects of mutations. Direct cell-cell contact or paracrine signaling produced by the tissue-specific microenvironment might allow the transformation of cells only with certain mutational events, forcing the selection of specific driver mutations in melanoma (*Pandiani et al., 2017*). Epidermal melanocytes interact mainly with keratinocytes, while internal melanocytes interact with various mesodermal stromas. Therefore, crosstalk between melanocytes and their cellular neighbors might prevent transformation or proliferation driven by specific signaling pathways.

In this paper, we investigated these possibilities. We forced the expression of oncogenic GNAQ$^{Q209L}$ in all melanocytes in mice beginning in adulthood using *Tyr-creER*. This drove the loss of melanocytes from the interfollicular epidermis (IFE) of the tail, while dramatically increasing melanocyte growth in the dermis. This observation demonstrated that GNAQ$^{Q209L}$ expression was not simply neutral for epidermal melanocytes; it was deleterious. We established primary cultures of normal or GNAQ$^{Q209L}$ expressing melanocytes sorted from the mouse tail epidermis and used time lapse microscopy to quantify melanocyte cell dynamics in the presence of either mouse embryonic fibroblasts (MEFs) or keratinocytes. We discovered that paracrine signaling from the epidermis reversibly switches GNAQ$^{Q209L}$ from an oncogene to an inhibitor of melanocyte survival and proliferation. Differential expression analysis of melanocytes sorted from the tail epidermis showed that the melanocytes expressing GNAQ$^{Q209L}$ exhibited alterations in gene expression related to cell adhesion, axon extension, oxidative stress, and apoptosis. Hence, our studies reveal the flip side of the G$\alpha_{q/11}$ signaling pathway in melanocytes.

## Results

### Forced GNAQ$^{Q209L}$ reduces melanocytes in mouse tail IFE

We previously generated a conditional GNAQ$^{Q209L}$ allele wherein oncogenic human GNAQ$^{Q209L}$ is expressed from the *Rosa26* locus in mice following the removal of a loxP-flanked stop cassette ('*R26-fs-GNAQ*$^{Q209L}$') (*Huang et al., 2015*). We previously showed that forcing GNAQ$^{Q209L}$ expression in melanocytes beginning during embryogenesis using the *Mitf-cre* transgene reduced the number of melanocytes in the IFE (*Huang et al., 2015*; *Alizadeh et al., 2008*). To test whether GNAQ$^{Q209L}$ affected melanocyte establishment or melanocyte maintenance, we used the tamoxifen (TM) inducible melanocyte-specific Cre transgene, tyrosinase *(Tyr)-creERT*$^2$ with *R26-fs-GNAQ*$^{Q209L}$ to induce GNAQ$^{Q209L}$ expression in melanocytes beginning in adulthood. We also included the *Rosa26-LoxP-Stop-LoxP-LacZ* reporter to label cells expressing GNAQ$^{Q209L}$ with LacZ (*Bosenberg et al., 2006*; *Soriano, 1999*). TM was administered once per day for 5 days at 4 weeks of age. One week following the last dose of TM, half of the mice were euthanized to collect and stain tail epidermal sheets with X-gal to determine the average number of melanocytes (LacZ-positive cells) per scale. At this time point, *R26-fs-GNAQ*$^{Q209L}$*/R26-fs-LacZ; Tyr-creERT*$^2$*/+* (hereafter referred to as 'GNAQ-lacZ') mice and *+/R26-fs-LacZ; Tyr-creERT*$^2$*/+* (hereafter referred to as 'WT-LacZ') mice showed no significant difference

in melanocyte numbers (p = 0.18, *Figure 1A and B*). However, 8 weeks following the last dose of TM, GNAQ-LacZ mice had on average 50% fewer melanocytes per scale than WT-LacZ mice (p = 0.045, *Figure 1A and B*). While scales in WT-LacZ mice were pigmented evenly, many scales in GNAQ-LacZ mice exhibited partial loss of melanin 8 weeks after TM (p = 0.048, *Figure 1C and D*). These results demonstrate that GNAQ$^{Q209L}$ signaling inhibits melanocyte maintenance in the IFE.

## The effects of GNAQ$^{Q209L}$ signaling in melanocytes are reversible

To isolate melanocytes from the epidermis for analysis in vitro, we used *Mitf-cre* with *Rosa26-LoxP-Stop-LoxP-tdTomato* and *Rosa26-fs-GNAQ$^{Q209L}$* to label the melanocyte lineage with a robust and intense fluorescent tdTomato signal. To confirm appropriate labeling, we first compared sections of the tail skin of *+/R26-fs-tdTomato; Mitf-cre/+* ('wildtype [WT]') and *R26-fs-GNAQ$^{Q209L}$/ R26-fs-tdTomato; Mitf-cre/+* ('GNAQ$^{Q209L}$') mice at 4 weeks of age (*Figure 1E–F*). Sections of WT tail skin exhibited tdTomato-positive cells in the IFE, located as expected on the basal membrane, as well as some tdTomato-positive cells in the dermis. In contrast, sections of GNAQ$^{Q209L}$ tail skin contained fewer tdTomato-positive cells in the IFE while exhibiting an abnormally extensive tdTomato signal in the dermis.

Next, we isolated tdTomato-positive melanocytes from the IFE of WT and GNAQ$^{Q209L}$ mice by fluorescent activated cell sorting (FACS) to study the melanocytes in in vitro settings. To obtain a single-cell suspension for FACS, we first split the tail skin IFE from the underlying dermis. Then, we dissociated the scales within the IFE with trypsin to obtain melanocytes and keratinocytes as dispersed single cells (Pop S, Urtatiz O, Van Raamsdonk CD. 2022. Isolation of inter-follicular epidermal melanocytes and keratinocytes from mouse tail skin. [Manuscript under Review at Bio-Protocol]). As expected, fewer tdTomato-positive cells were sorted from GNAQ$^{Q209L}$ mice (0.61%) compared to WT mice (0.99%) (p = 0.012, *Figure 2A and B*). Previous studies have estimated that melanocytes account for ~1.5% of the total cells in the IFE (*Glover et al., 2015*). Note, since some melanocytes have been lost in the GNAQ$^{Q209L}$ IFE, this approach may have selected for melanocytes that are more resistant to the effects of GNAQ$^{Q209L}$.

We then seeded the sorted IFE melanocytes on fibronectin-coated plates to study baseline survival. The GNAQ$^{Q209L}$ expressing melanocytes survived better than the WT melanocytes (p = 3.3 × 10$^{-5}$ for genotype, two-way ANOVA analysis, *Figure 2C*). Hence, the growth-inhibiting effects of GNAQ$^{Q209L}$ expression in IFE melanocytes are reversible. When removed from the epidermis and grown on fibronectin, IFE melanocyte survival was increased by GNAQ$^{Q209L}$ expression.

To compare the effects of GNAQ$^{Q209L}$ with another well-studied melanoma oncogene, we used the same methods to obtain BRAF$^{V600E}$ expressing melanocytes from the mouse tail IFE. In mice, the expression of BRAF$^{V600E}$ leads to melanocytic over-growth in both the tail dermis and epidermis (*Urtatiz et al., 2018*). We crossed the conditional *Braf$^{V600E}$* allele, which expresses *Braf$^{V600E}$* from the endogenous *Braf* locus (*Dankort et al., 2007*) to *R26-fs-tdTomato* and *Mitf-cre*. Then, we used FACS to isolate epidermal melanocytes from the tail IFE of 4-week-old mice (*+/R26-fs-tdTomato; Mitf-cre/+; Braf$^{CA}$/+*, hereafter referred to as 'BRAF$^{V600E}$'). We found that BRAF$^{V600E}$ expression increased the survival of sorted IFE melanocytes plated on fibronectin, even more so than GNAQ$^{Q209L}$ (*Figure 2C*). Melanocytes expressing BRAF$^{V600E}$ could be maintained in culture for at least 8 days (*Figure 2G*).

We observed a striking difference in cell morphology of GNAQ$^{Q209L}$ melanocytes compared to WT melanocytes grown in vitro. GNAQ$^{Q209L}$ melanocytes exhibited a more dendritic morphology with several protrusions per cell, while WT melanocytes appeared mostly spindle-shaped (*Figure 2D*). To study the cellular dynamics of the melanocytes, we analyzed time lapse microscopy for 20 hr (*Figure 2E and F*). No migration, formation of new protrusions, or cell divisions were observed in the melanocytes of either the WT or GNAQ$^{Q209L}$ cultures plated on fibronectin alone. However, in WT cultures, some melanocytes were observed to adopt a round shape, after which the cells were immediately lost from view (presumed cell death, example shown in the circled cell in *Figure 2E*).

As another test of GNAQ$^{Q209L}$ reversibility, we isolated MEFs and plated them onto fibronectin-coated plates. Then we added FACS sorted IFE melanocytes from either WT or GNAQ$^{Q209L}$ mouse tails. The growth of both WT and GNAQ$^{Q209L}$ expressing melanocytes was stimulated by MEF co-culture, as there was an increase in cell number above the original number plated, unlike on fibronectin alone (*Figure 2H*). In this situation, we did not detect a difference in survival between WT and GNAQ$^{Q209L}$ melanocytes (p = 0.48, two-way ANOVA). This was surprising, since GNAQ$^{Q209L}$ stimulates melanocyte

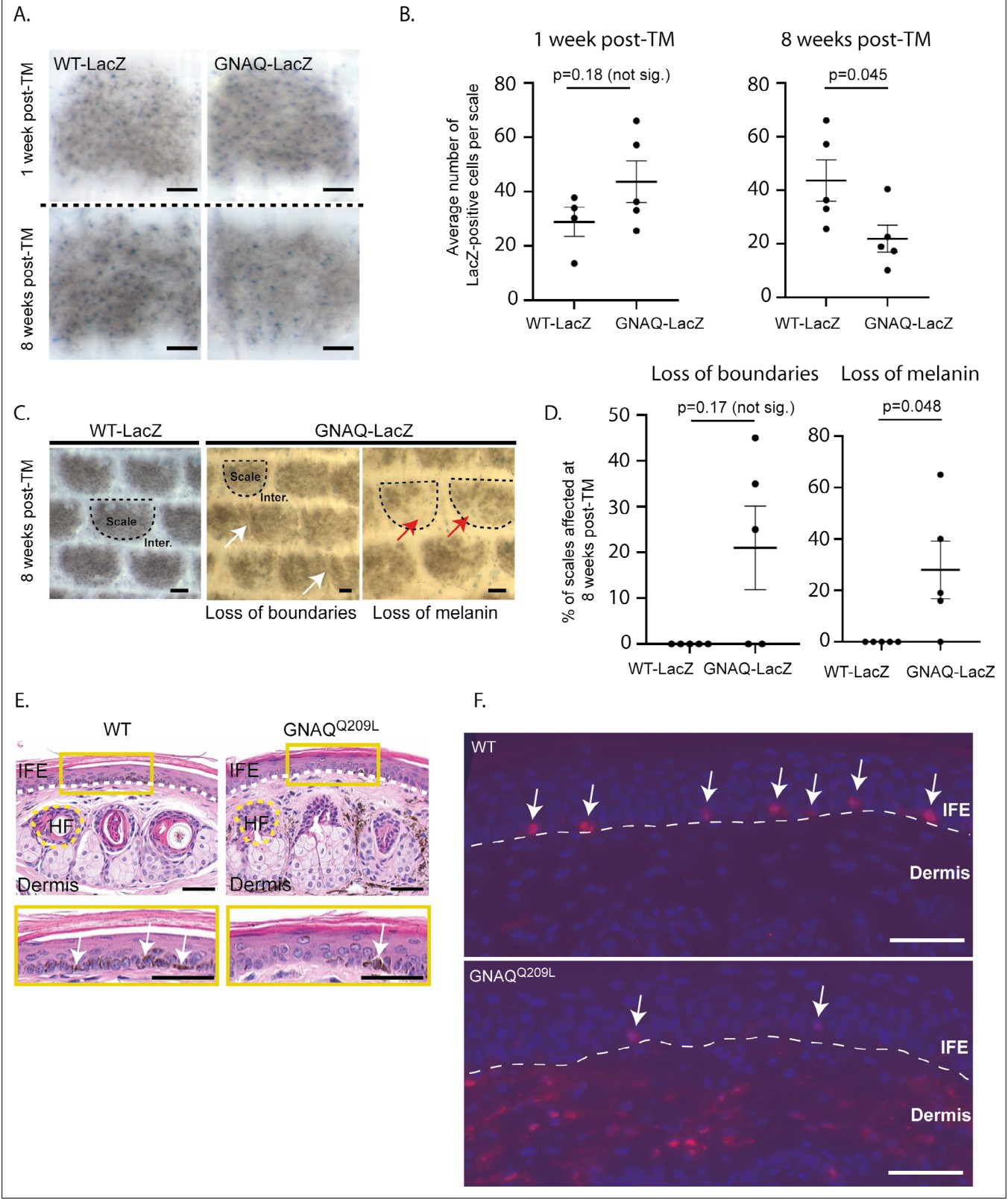

**Figure 1.** Forced GNAQ^(Q209L) signaling reduces the number of melanocytes in the interfollicular epidermis (IFE). (**A**) Representative example of LacZ-positive cells within scales of WT-LacZ and GNAQ -LacZ tails at 1 or 8 weeks post-tamoxifen (TM) treatment. (**B**) Quantification of the average number of LacZ-positive cells per scale at 1 and 8 weeks post-TM treatment. (Each point represents average for one mouse; mean ± SEM; unpaired t-test.) (**C**) Representative examples of X-gal stained whole mount epidermal tail sheets in WT-LacZ and GNAQ-LacZ mice at 8 weeks post-TM, showing loss of

*Figure 1 continued on next page*

*Figure 1 continued*

boundaries in scale pigmentation (white arrows) or loss of melanin (red arrows) in GNAQ-LacZ mice. Example scales are outlined in dashed line for reference. (**D**) Percentage of epidermal scales exhibiting loss of boundaries or loss of melanin in five WT-LacZ and five GNAQ-LacZ mice at 8 weeks post-TM. (Each point represents % for one mouse; Kolmogorov-Smirnov test, mean ± SEM.) (**E**) H&E stained cross sections of tail skin in wildtype (WT) and GNAQ$^{Q209L}$ mice. The yellow box below shows a magnified area of the interfollicular epidermis (IFE). Less melanin was observed in the IFE of GNAQ$^{Q209L}$ skin (white arrows point to examples). Dashed lines indicate the boundaries between the IFE, dermis, and an example hair follicle (HF). (**F**) tdTomato expression (red) in cross sections of tail skin of WT and GNAQ$^{Q209L}$ mice at 4 weeks of age showing a reduced number of melanocytes (tdTomato+ cells) in the IFE of GNAQ$^{Q209L}$ mice and an abnormal expansion of melanocytes in the dermis. Sections are counterstained with DAPI (blue). Dashed lines indicate the boundaries between the IFE and dermis. White arrows indicate melanocytes located in the IFE. In A and C, scale bars represent 100 μm, while in E and F, scale bars represent 50 μm.

hypertrophy in the dermis (*Figure 1E and F*). This suggests that MEFs do not adequately simulate a dermis-like environment. However, when grown with MEFs, GNAQ$^{Q209L}$ melanocytes grew much larger and irregularly shaped compared to WT melanocytes (*Figure 2I*).

To summarize, our findings suggest that the attrition of GNAQ$^{Q209L}$ expressing IFE melanocytes from the mouse tail epidermis is not due to an inherent characteristic in IFE melanocytes because the phenotype is reversible by switching the microenvironment in vitro.

## The IFE impairs the survival and proliferation of GNAQ$^{Q209L}$ melanocytes in vitro

Keratinocytes are known to affect melanocyte proliferation and survival in the IFE in a normal context, so we suspected that interactions between keratinocytes and melanocytes in the IFE could be modifying the outcome of GNAQ$^{Q209L}$ signaling. To test this, we dissociated tail IFE into single cells and plated the suspension onto fibronectin-coated plates. Co-culturing with dispersed IFE provided a significant boost to WT melanocyte survival compared to fibronectin coating alone (p = 0.0071 for culture condition, two-way ANOVA, *Figure 3A*). In contrast, the co-culture of IFE with GNAQ$^{Q209L}$ expressing melanocytes caused a significant reduction in melanocyte survival throughout the 5 days of the experiment (p = 0.0053 for culture condition, two-way ANOVA, *Figure 3B*).

Observation using time lapse microscopy showed that both WT and GNAQ$^{Q209L}$ melanocytes exhibited different behavior when cultured with IFE compared to culture on fibronectin alone, with both WT and GNAQ$^{Q209L}$ melanocytes migrating and extending and retracting protrusions (*Figure 3C*, example *Videos 1 and 2*). GNAQ$^{Q209L}$ expressing melanocytes exhibited longer and more numerous protrusions (p = 0.0096 and p = 0.00041, respectively, *Figure 3E and F*). Perhaps related to the unusually long protrusion length, breakage and fragmentation of dendrites was observed in 45% of the GNAQ$^{Q209L}$ melanocytes tracked by time lapse microscopy for 20 hr (*Figure 3H and I*). In comparison, this phenomenon was observed in just 5% of WT melanocytes (p = 0.021). The average cell area of GNAQ$^{Q209L}$ melanocytes was 2.7-fold greater than WT melanocytes (p = $2.6 \times 10^{-13}$, *Figure 3G*) and the cells were less circular (p = $5.0 \times 10^{-8}$, *Figure 3D*). These changes in cell morphology and dynamics did not seem to affect cell migration (*Figure 4A*). We found no difference in the total length traveled or the straightness of path ("Directness") in GNAQ$^{Q209L}$ versus WT melanocytes cultured with IFE (p = 0.86 and p = 0.18, *Figure 4B and C*).

We also noticed that there was cell division in the time lapse microscopy images. During the 20 hr of observation, 16% of WT melanocytes co-cultured with IFE underwent a cell division event (*Figure 4D*, example shown in *Figure 4E*, *Video 1*). WT melanocytes smoothly progressed from cleavage furrow formation to the separation of daughter cells and on average this took about 50 min (*Figure 4H*). In contrast, only one of the 99 tracked GNAQ$^{Q209L}$ melanocytes divided (p = 0.0040, *Figure 4D*). In this cell, furrow formation was not as clear and the process took 150 min (*Figure 4F*). We used the same methods to examine the effects of the BRAF$^{V600E}$ oncogene on proliferation (*Figure 4G*). During 20 hr of observation, 32% of BRAF$^{V600E}$ melanocytes underwent a cell division event (*Figure 4D*). Cell division took the least time in Braf$^{V600E}$ expressing melanocytes (*Figure 4H*).

The observation of cell division in WT cells in the time lapse images was unexpected given that these cells did not increase above the number plated in the survival curves in *Figure 3A*. We think the difference is due to the culture conditions in the survival curve experiments versus the time lapse imaging experiments. In the experiments in *Figure 3A*, there was less media relative to cells than there was in the experiments in *Figure 4*. This was because cells in *Figure 3A* were cultured in a 96-well

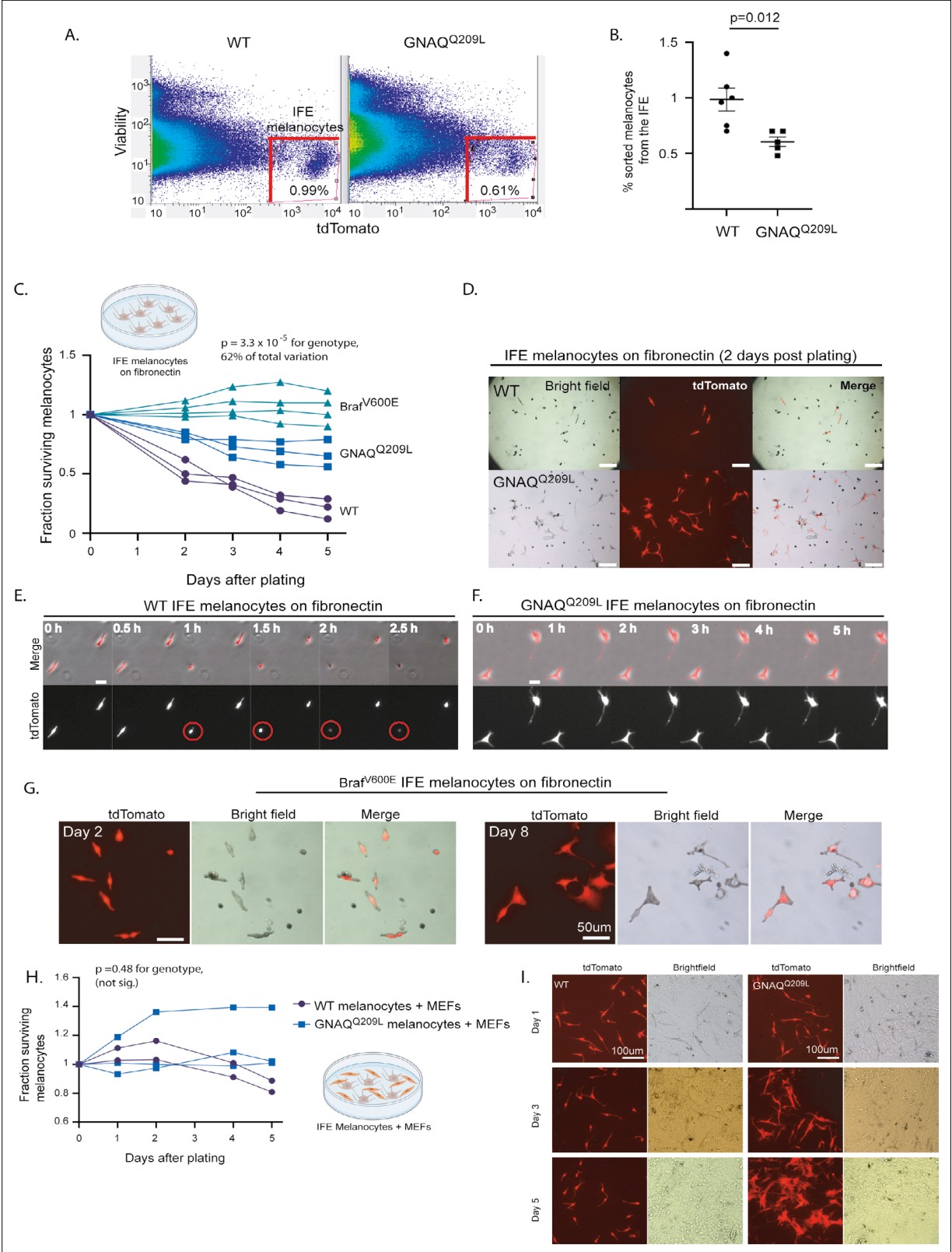

**Figure 2.** Fluorescent activated cell sorting (FACS) sorted GNAQ^Q209L and BRAF^V600E interfollicular epidermis (IFE) melanocytes cultured on fibronectin have increased survival compared to wildtype (WT). (**A**) Representative examples of FACS dot plots of single-cell suspension from the IFE epidermis of WT and GNAQ^Q209L mice. (**B**) Average percentage of tdTomato+ cells sorted from the IFE of WT and GNAQ^Q209L mice, with a significantly smaller percentage in GNAQ^Q209L. (Each point represents % from one mouse, mean ± SEM; unpaired t test). (**C**) Fraction of surviving WT, GNAQ^Q209L, and

*Figure 2 continued on next page*

*Figure 2 continued*

Braf[V600E] FACS sorted melanocytes plated on fibronectin (each line represents an independently derived primary culture, two-way ANOVA). (**D**) Representative images of melanocyte morphology 2 days post-plating on fibronectin-coated wells, showing increased dendrite formation in GNAQ[Q209L] melanocytes. (**E**) Time lapse microscopy of two WT melanocytes plated on fibronectin. The circled cell adopted a round shape shortly before being lost from view. (**F**) Time lapse microscopy of two GNAQ[Q209L] melanocytes showing a dendritic cell morphology that remained stable over time. (**G**) Representative images of BRAF[V600E] melanocytes at 2 and 8 days post-plating on fibronectin. Scale bars represent 100 μm in D, 20 μm in E and F, and 50 μm in G. (**H**) Fraction of surviving WT and GNAQ[Q209L] IFE melanocytes co-cultured with mouse embryonic fibroblasts (MEFs), showing initial growth above the baseline for both genotypes before loss began (each line represents an independently derived primary culture, two-way ANOVA). (**I**) Representative images of FACS sorted WT and GNAQ[Q209L] IFE melanocytes co-cultured with MEFs. While the WT and GNAQ[Q209L] IFE melanocytes had a similar spindle cell morphology at day 1, the GNAQ[Q209L] melanocytes progressively developed large and abnormal shapes.

plate format, while the cells in *Figure 4* were cultured in 35 mm dishes on coverslips. However, under both set-ups, WT melanocytes did better than GNAQ[Q209L] melanocytes. Interestingly, the different set-ups show that both proliferation and survival can be impacted by GNAQ[Q209L].

To determine whether the reduced survival of GNAQ[Q209L] expressing melanocytes co-cultured with IFE was due to direct contact inhibition or a paracrine mechanism, we used a transwell assay in which we seeded FACS collected IFE melanocytes on a permeable membrane with dissociated IFE from the same animal plated in the well underneath, in the 96-well format. If direct cell-cell contact was necessary for the inhibitory effect of the IFE, then we would expect increased survival of GNAQ[Q209L] melanocytes in this culture system compared to simple co-culture. We found that it made no difference whether the IFE was directly in contact with the GNAQ[Q209L] expressing melanocytes or present in the same dish, separated by a membrane (*Figure 5B and C*). In WT melanocytes, there was no significant difference in survival for the first 3 days, after which the transwell melanocytes developed an advantage (*Figure 5A and C*). We conclude that paracrine signaling from the microenvironment helps switch GNAQ[Q209L] from an oncogene to an inhibitor of cell survival.

Altogether, the cell culture experiments revealed that IFE co-culture stimulates survival and cell division of WT melanocytes, but inhibits these processes in GNAQ[Q209L] melanocytes. Furthermore, the IFE stimulated protrusion activity in both WT and GNAQ[Q209L] cells, with the GNAQ[Q209L] expressing cell hyper-responding. Since both WT and GNAQ[Q209L] melanocytes were static when plated on fibronectin alone, these experiments show that the IFE plays a vital role in melanocyte regulation. We conclude that crosstalk between melanocytes and the IFE reversibly switches the outcome of GNAQ signaling from promoting to inhibiting melanocyte growth and survival.

## Gene expression analysis: regulation of cell adhesion and pseudopod dynamics

To identify the pathways that change in response to GNAQ[Q209L] expression in IFE melanocytes, we performed RNA sequencing (RNAseq) immediately after sorting WT and GNAQ[Q209L] melanocytes from tail IFE (n = 6 mice and n = 3 libraries per genotype). There were 14,461 genes with an average FPKM >0.1 in WT and/or GNAQ[Q209L] melanocytes (*Supplementary file 1a*). Seven genes previously shown to be directly involved in pigment production and melanosome biology were on the list of the top 20 most highly expressed genes in WT melanocytes, validating our melanocyte isolation protocol (*Supplementary file 2a*). These genes were *Pmel*, *Dct*, *Tyrp1*, *Mlana*, *Cd63*, *Slc24a5,* and *Gpnmb*. We analyzed differential gene expression using DEseq2; 1745 genes were found to be differentially expressed (DE), of which 729 genes were down-regulated and 1016 genes were up-regulated in GNAQ[Q209L] melanocytes (cutoff for q-value <0.05) (*Supplementary file 1b and c*).

We first used the DAVID Functional Annotation Tool (6.8) to understand the biological significance of the DE genes in GNAQ[Q209L] melanocytes. We analyzed the combined list of up- and down-regulated DE genes with a $\log_2$ fold change ('LFC') of >2.0 or < –2.0. Using Gene Ontology (GO) analysis and KEGG pathway analysis, there were two dominant signatures. One overlapping set of genes supported the terms: cell adhesion, focal adhesion, extracellular matrix-receptor interactions, and extracellular matrix structural constituents (*Figure 6A*, *Supplementary file 1d*). Overlapping genes supported a second signature for the terms: axons, axon guidance, and nervous system development (*Supplementary file 1e*).

For changes related to cellular adhesion, almost all the identified genes were up-regulated in expression. This included types *IV*, *V*, and *XXVII collagens*, *Fn1* (Fibronectin 1), *Itga1* (Integrin alpha

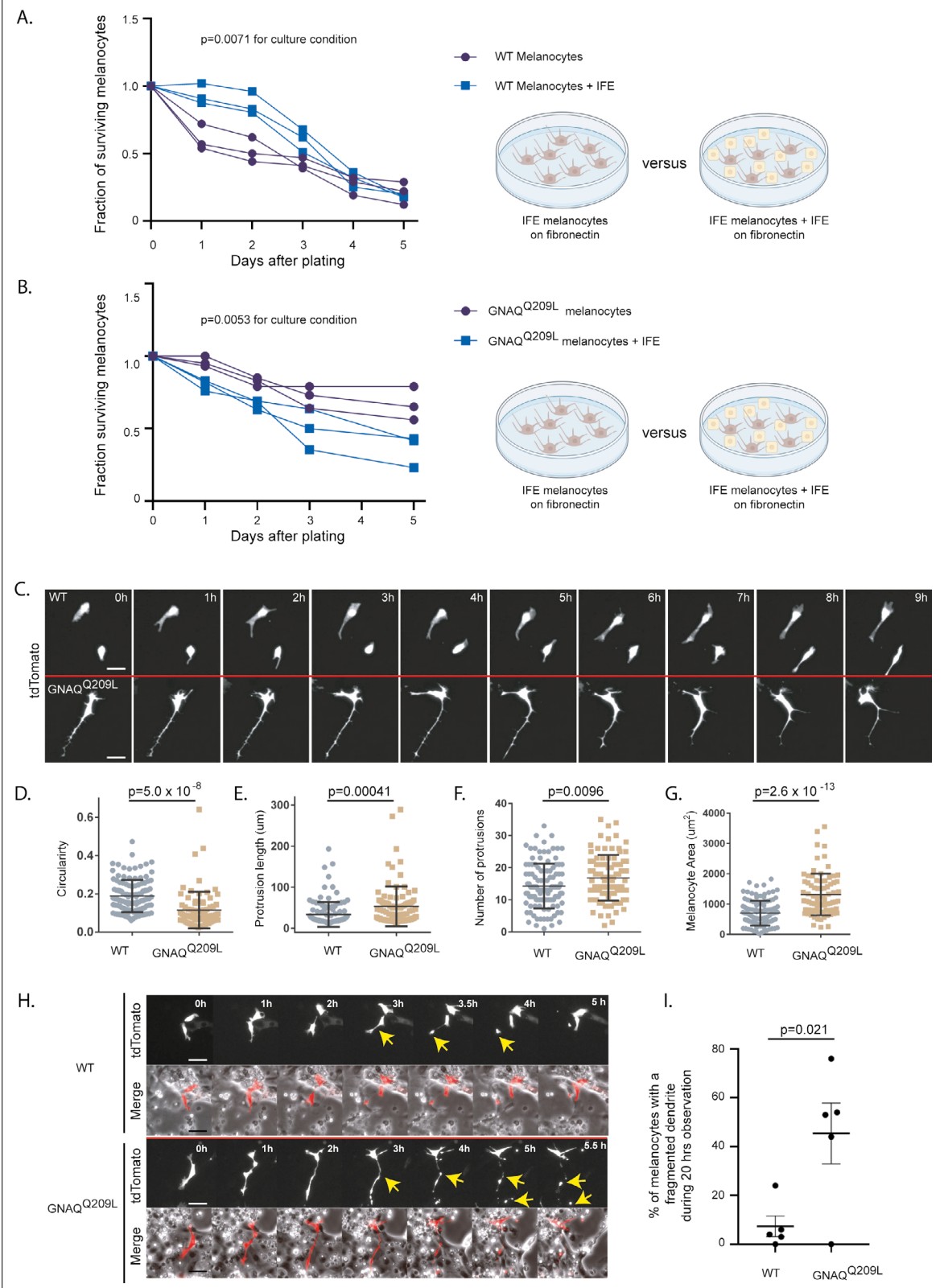

**Figure 3.** The interfollicular epidermis (IFE) impairs survival and alters pseudopod dynamics in GNAQ$^{Q209L}$ melanocytes. (**A**) Survival of unsorted wildtype (WT) melanocytes plated with its IFE, compared to sorted IFE WT melanocytes plated onto fibronectin. The presence of IFE increased the survival of WT melanocytes. (Each line represents one independently derived culture, two-way ANOVA.) (**B**) Survival of unsorted GNAQ$^{Q209L}$ melanocytes plated with its IFE, compared to sorted IFE GNAQ$^{Q209L}$ melanocytes plated onto fibronectin. The presence of IFE decreased the survival of GNAQ$^{Q209L}$ melanocytes.

*Figure 3 continued on next page*

*Figure 3 continued*

(Each line represents one independently derived culture, two-way ANOVA.) (**C**) Time lapse images show representative WT and GNAQ^Q209L melanocytes co-cultured with IFE between 0 and 8 hr. The GNAQ^Q209L cell exhibits abnormally long dendrites and a less circular (more polygonal) cell body shape. (**D–G**) Quantification of circularity (**D**), protrusion length (**E**), number of protrusions (**F**), and melanocyte area (**G**), in WT and GNAQ^Q209L melanocytes co-cultured with IFE. (Each point represents the measurement of one cell, mean ± SEM; unpaired t test.) (**H**) Time lapse microscopy showing a representative example of dendrite fragmentation in WT and GNAQ^Q209L melanocytes. Arrows indicate dendrite breakage points and subsequent fragments that form up into balls. (**I**) Quantification of the percent of cells experiencing dendrite fragmentation in WT and GNAQ^Q209L melanocytes cultured with IFE. (Each point represents the measurement from one culture, mean ± SEM; unpaired t test.) Scale bars represent 40 μm in C and H.

1), *Itgb3* (Integrin beta 3), *Amigo1*, *Mcam*, *Cadherin 6*, *Cercam*, and *Esam*. Examining the complete list of DE genes (with no cut-off), we also found that two Rho GEFs that interact with $G\alpha_{q/11}$, *Trio* and *Kalrn* (Kalirin), were up-regulated (LFC 0.5 and 1.8). Rho GEFs catalyze GDP to GTP exchange on small Rho guanine nucleotide-binding proteins, regulating the actin cytoskeleton. We also noticed that several genes that play an essential role in lamellipodia formation (*Ridley, 2015*) including *Rhob*, *Rhoc*, *Cdc42*, and *Rock* were all up-regulated by LFC > 1.0. Increased adhesion and cell-ECM interactions could underlie the changes in cell shape and increased cell area in GNAQ^Q209L melanocytes co-cultured with IFE (*Figure 3*).

To examine the actin cytoskeleton in GNAQ^Q209L and WT melanocytes, we measured the overall level of actin alignment, termed cell fibrousness, using an in vitro quantitative approach previously described by *Haage et al., 2018*. First, WT and GNAQ^Q209L tdTomato-positive melanocytes co-cultured with IFE were stained for f-actin using phalloidin (*Figure 6B*). Then, confocal z-projections of individuals melanocytes were obtained and processed using quantitative image analysis to determine cell fibrousness. GNAQ^Q209L expressing melanocytes exhibited less cell fibrousness than WT melanocytes ($p = 1.8 \times 10^{-12}$, *Figure 6C*). Hence, GNAQ^Q209L expressing melanocytes have less organized actin cytoskeletons.

Melanocytes are known to share signaling pathways with neurons (*Yaar and Park, 2012*). The DE genes related to axons (identified by pathway analysis in *Supplementary file 1e*) are diverse in function; however, the number of Semaphorin genes on the list stands out, suggesting that these signaling molecules might be important. Among the complete list of DE genes, the Semaphorins that were up-regulated were *Sema3c*, *Sema3d*, *Sema3g*, *Sema4c*, and *Sema4f*, while *Sema3a*, *Sema3b*, *Sema4b*, *Sema5a*, *Sema6d*, and *Sema7a* were down-regulated. In GNAQ^Q209L melanocytes, the expression of *Itgb3* (Integrin beta 3) was also up-regulated (LFC 3.6), while *Plexc1* (Plexin C1) was down-regulated (LFC –0.7). These expression changes in *Sema3a*, *Sema7a*, *Itgb3*, and *Plexc1* are consistent with the abnormally long dendrites and increased cellular area in GNAQ^Q209L expressing melanocytes, based on previous results in neurons (*Scott et al., 2008*;

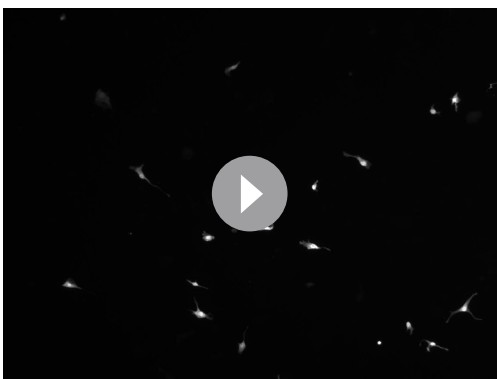

**Video 1.** Wildtype (WT) melanocytes co-cultured with interfollicular epidermis (IFE), 625 min time lapse. Cell protrusions, cell movements, and cell division events can be seen in the melanocytes in this video. Cell protrusions are shorter in WT cells than in GNAQ^Q209L. One of the cell division events in this video occurs at around 9 s in a cell near the center of the field.

https://elifesciences.org/articles/71825/figures#video1

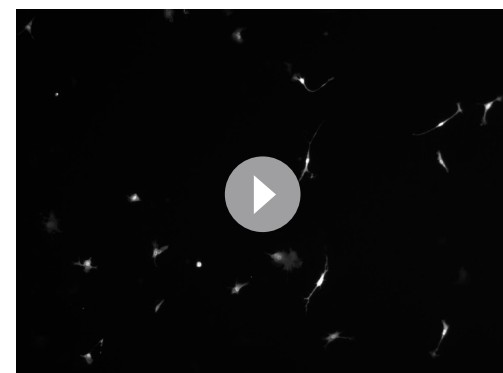

**Video 2.** GNAQ^Q209L melanocytes co-cultured with interfollicular epidermis (IFE), 625 min time lapse. Cell protrusions and cell movements can be seen in the melanocytes in this video. Protrusions are longer in GNAQ^Q209L melanocytes than in wildtype (WT) and some cells take on usual shapes. The bottom right cell shows an example of dendrite breakage, with the fragmented piece balling up and drifting away.

https://elifesciences.org/articles/71825/figures#video2

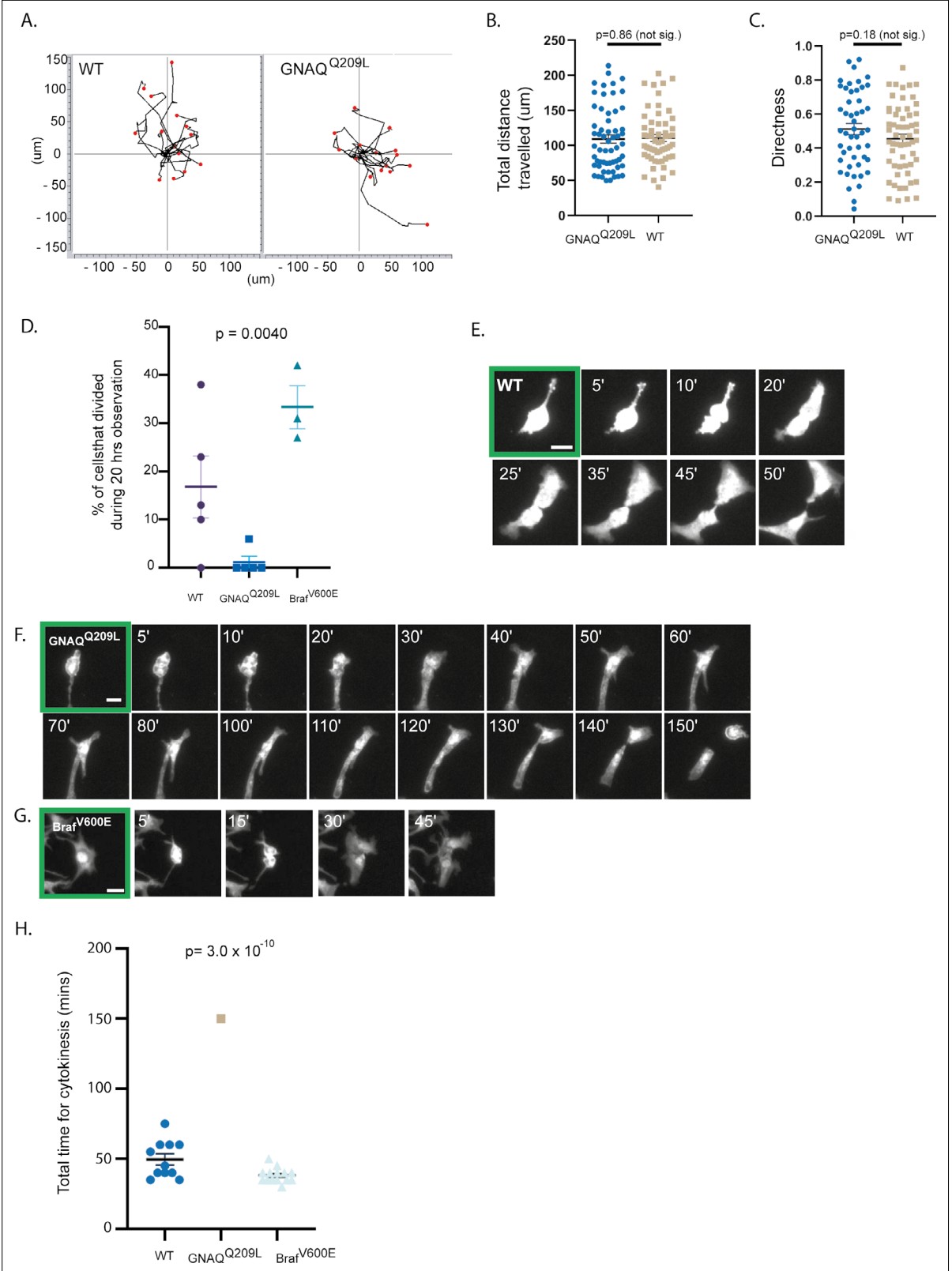

**Figure 4.** The interfollicular epidermis (IFE) impairs cell division of GNAQ^Q209L melanocytes. (**A**) Migration plots of wildtype (WT) and GNAQ^Q209L melanocytes co-cultured with IFE over 20 hr. (**B**) Quantification of total distance traveled in 20 hr. (Each point represents the measurement from one cell, mean ± SEM; unpaired t test.) (**C**) Quantification of the directness of cell trajectories over 20 hr. Directness = 1 is a straight line cell trajectory. (Each point represents the measurement from one cell, mean ± SEM; unpaired t test.) (**D**) Percentage of tracked cells undergoing division during 20 hr of time

*Figure 4 continued on next page*

*Figure 4 continued*

lapse microscopy, when co-cultured with IFE. (Each point represents the measurement from one culture, mean ± SEM; ordinary one-way ANOVA.) (**E–G**) Representative examples of cell division events from cleavage furrow formation to separation of daughter cells in a WT melanocyte (in E), a GNAQ[Q209L] melanocyte (in F), and a BRAF[V600E] melanocyte (in G). (**H**) Quantification of the time taken between cleavage furrow formation to separation into daughter cells in melanocytes co-cultured with IFE. (Each point represents the measurement from one cell, mean ± SEM, ordinary one-way ANOVA.) Scale bar in E–G represents 20 µm.

*Mann et al., 2007*; *Schwarz et al., 2008*). To summarize, the RNAseq data suggests that the impact of GNAQ[Q209L] expression on IFE melanocyte morphology is mediated through increased cell adhesion, a disorganized actin cytoskeleton and the promotion of dendrite extensions.

## Gene expression analysis: cellular stress and apoptosis

Using the RNAseq data, gene set enrichment analysis (GSEA) revealed that GNAQ[Q209L] IFE melanocytes express a pattern of genes related to cellular stress and apoptosis. GSEA showed that GNAQ[Q209L] melanocytes are enriched in the hallmarks for 'apoptosis', 'p53 pathway', and 'hypoxia' (*Figure 6D*). Among the top-ranked DE genes was *Stc1* (Stanniocalcin) which encodes a glycoprotein hormone involved in calcium/phosphate homeostasis. *Stc1* was up-regulated by an LFC of 7.1 in GNAQ[Q209L] melanocytes. Significant up-regulation of *STC1* has been reported in tumors under hypoxic or oxidative stress (*Yeung et al., 2005*; *Nguyen et al., 2009*; *Guo et al., 2013*; *Block et al., 2009*; *Lai et al., 2007*). *Cdkn2a* and *Ccnd1* were up-regulated in GNAQ[Q209L] expressing melanocytes by an LFC of 5 and 1.7, respectively, similar to the intracellular response to oxidative stress previously described in melanocytes in the disease, vitiligo, which results in the loss of melanocytes from the epidermis (*Bellei et al., 2013*; *Becatti et al., 2014*; *Choi et al., 2010*). *S100b*, which encodes a negative regulator of P53, was down-regulated by LFC –2.6. To assess the amount of cell death in vivo, we dissociated IFE from mouse tails and stained the cells using fluorescently tagged Annexin-V, which binds to phosphatidylserine abnormally present in the outer leaflet of the cell membrane in early apoptotic cells. Using FACS, we quantified the percentage of tdTomato and Annexin-V double-positive cells (*Figure 6E*). There was a greater percentage of Annexin-V-positive melanocytes from GNAQ[Q209L] tail epidermis (4.2% versus 1.2%). This supported the GSEA data that suggests that the loss of GNAQ[Q209L] expressing melanocytes from the IFE involves apoptosis.

## Gene expression analysis: signaling pathways downstream of Gα$_q$

In uveal melanoma, activation of Gα$_{q/11}$ drives cell proliferation and stimulates the 'MAPK' pathway (*Chen et al., 2017*) and activates the Hippo pathway through nuclear localization of YAP1 via a TRIO-RHO/RAC signaling circuit (*Feng et al., 2014*). Neither of these pathways was identified by the pathway analysis (above), so we took a more focused examination of the genes expected to read out MAPK and Hippo pathway activity. Wagel et al. identified a 10 gene set to score MAPK pathway activity across various human cancer types (*Wagle et al., 2018*). The genes are *SPRY2*, *SPRY4*, *ETV4*, *ETV5*, *DUSP4*, *DUSP6*, *CCND1*, *EPHA2*, *EPHA4*, and *PHLDA1*. *INPP5F*, *MAP2K3*, *TRIB2*, and *ETV1* were additional MAPK targets (*Wagle et al., 2018*). Half of these MAPK target genes were significantly up-regulated in the GNAQ[Q209L] IFE melanocytes (*Spry2*, *Spry4*, *Etv5*, *Ccnd1*, *Inpp5f*, *Map2k3*, and *Trib2*.) Two genes were down-regulated (*Dusp4* and *Phlda1*) and the rest were unchanged (*Supplementary file 1ab and c*). On the balance, this suggests that the MAPK pathway is up-regulated, although perhaps not maximally.

In contrast, there was no significant up-regulation of any of the genes previously associated with TEAD activation downstream of the non-canonical Hippo pathway in uveal melanoma by *Li et al., 2019* (*Ctgf*, *Cyr611*, *Ankrd1*, *Pkrcd*, *Nras*, *Rras2*, *Rasgrp1*, and *Rasgrp3*). While *Trio* was slightly up-regulated (LFC 0.5), *Rac1* and *Rhoa* were not DE.

To further investigate, we checked PANTHER pathways (http://geneontology.org/) with our subset of DE genes with >2 or <–2 fold change. Interestingly, this analysis hit just two pathways (both up-regulated): the 'heterotrimeric G protein signaling pathway-Gq alpha and Go alpha mediated pathway' (FDR adjusted p = 5.23 × 10$^{-3}$) and the 'integrin signaling pathway' (FDR adjusted p = 1.4 × 10$^{-3}$). For the G protein pathway, four of the eight genes supporting the hit were members of the regulator of G protein signaling family (*Rgs2*, *Rgs3*, *Rgs9*, and *Rgs16*). These genes were up-regulated (LFC 2.2–3.7) in the GNAQ[Q209L] melanocytes, which would be expected to decrease Gα$_{q/11}$ signaling, perhaps as an

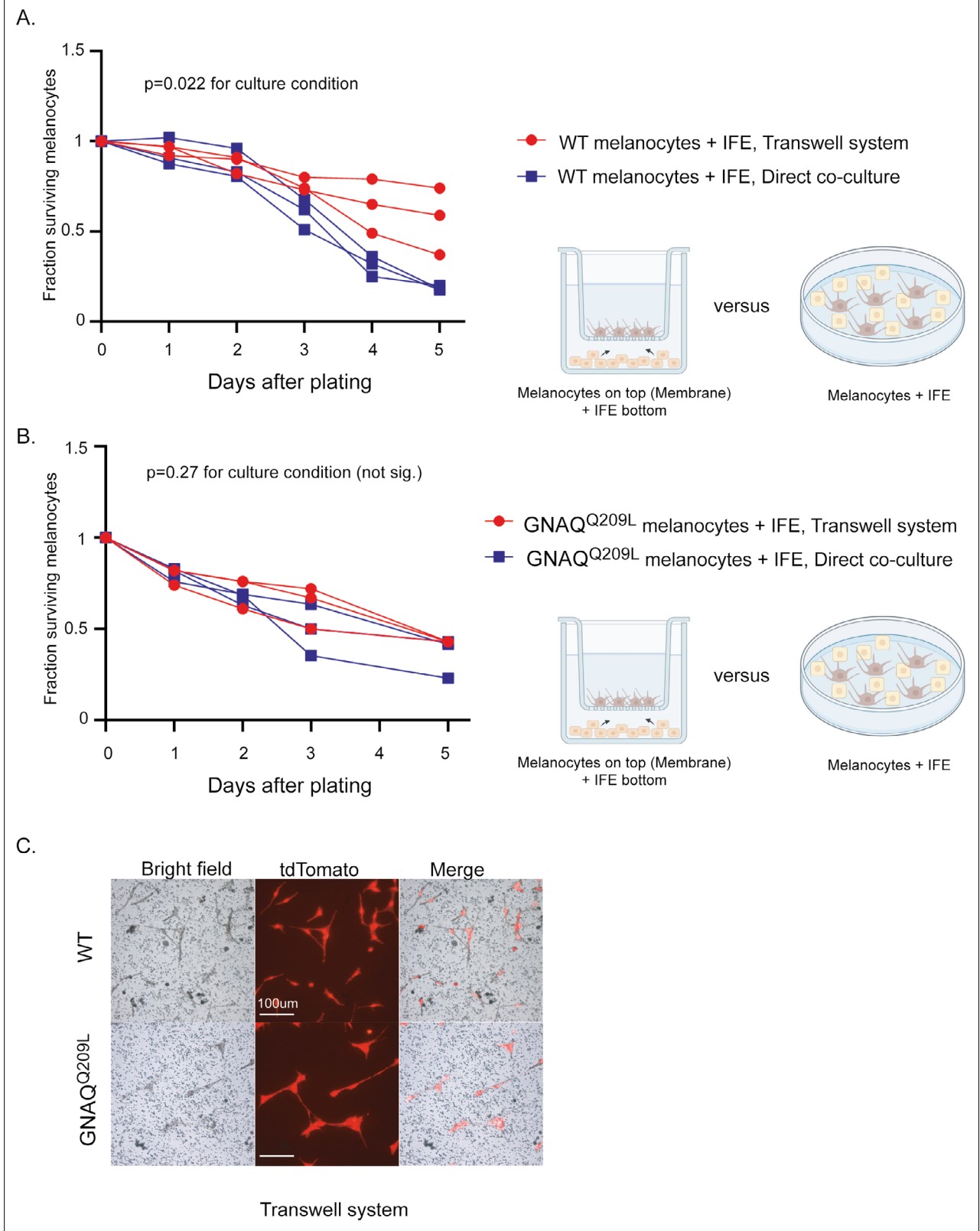

**Figure 5.** Interfollicular epidermis (IFE) control of melanocytes was maintained in a transwell culture system. (**A**) Comparison of survival for wildtype (WT) melanocytes in direct contact with IFE ('direct co-culture') versus a transwell system where the two populations are separated by a permeable membrane. In WT melanocytes, there was no significant difference in survival for the first 3 days, after which the transwell melanocytes developed an advantage. (Each line represents one independent culture, mean ± SEM; two-way ANOVA.) (**B**) Comparison of survival for GNAQ^Q209L melanocytes as in

*Figure 5 continued on next page*

*Figure 5 continued*

A. There was no significant difference in survival throughout the culture period. (Each line represents one independent culture, mean ± SEM; two-way ANOVA.) (**C**) Representative images of WT and GNAQ^Q209L melanocytes at day 3 on the transwell membrane (no direct cell contact with IFE). GNAQ^Q209L melanocytes were larger than WT melanocytes, as in direct co-culture. Scale bar represents 100 µm in C.

attempt to offset the constitutively active GNAQ^Q209L signal (*Hurst and Hooks, 2009*). The other four supporting genes for the G protein pathway were *Sdk1* (Sidekick cell adhesion molecule 1), which was down-regulated, and *Adrbk2* (G protein-coupled receptor kinase 3), *Gng4* (Guanine nucleotide binding protein gamma 4), and *Cacna1a* (Calcium voltage-gated channel subunit alpha1 A), which were all up-regulated.

## Low frequency of *GNAQ* and *GNA11* oncogenic mutations in human cutaneous melanoma

Our results suggested that oncogenic mutations in *GNAQ* and its closely related homolog, *GNA11*, are rarely found in cutaneous melanoma because the epidermal microenvironment causes Gα$_{q/11}$ signaling to inhibit melanocytes. To determine the frequency of these mutations, we searched the COSMIC database to identify all cutaneous melanoma cases with a *GNAQ* or *GNA11* oncogenic mutation at either Q209 or R183; 2753 and 2295 samples meeting our inclusion criteria (see legend of *Supplementary file 2b and c*) had *GNAQ* or *GNA11* mutation status reported, respectively. A total of 23 cases carried an oncogenic mutation in either *GNAQ* (n = 11) or *GNA11* (n = 12). Five of these cases were actually specified as malignant blue nevus or uveal melanoma in their original publications and so were not cutaneous melanomas. Eight other cases carried one or more mutations in genes also found in uveal (*Harbour et al., 2010*; *Martin et al., 2013*) or mucosal (*Hayward et al., 2017*; *Sheng et al., 2016*) melanoma, but not in cutaneous melanoma: *BAP1*, *SF3B1*, or *EIF1AX* and hence were similarly suspect as misclassified. One case was from a cell line and in another case, the frequency of an odd *GNAQ^Q209H* mutation in the tumor was less than 5%. This left eight cases that we think could have arisen in the epidermis, for an incidence of 0.15% (4/2753) for *GNAQ* and 0.17% (4/2295) for *GNA11*. Interestingly, three of the eight cases carried a mutation in either *BRAF* (V600E) or *NRAS* (Q61H or Q61R), suggesting that these oncogenes could offset or over-ride the growth inhibitory effects of constitutively active Gα$_{q/11}$ signaling. Another case carried a *P53^Y220C* mutation, which would be expected to inhibit apoptosis (*Supplementary file 2b and c*).

## Discussion

Oncogenic *GNAQ* and *GNA11* mutations are observed at a high frequency in human melanocytic neoplasms in non-epithelial tissues. Still, they are rare in anatomical locations with an epithelial component, such as the IFE or conjunctiva of the eye. One theoretical explanation for this skewed frequency is that some unknown mechanism mutates *GNAQ* and *GNA11* only in non-epithelial locations. We found that this is not the case. The forced expression of oncogenic GNAQ^Q209L in mature melanocytes in mice did not result in cutaneous melanoma, but instead had the opposite effect: the gradual loss of melanocytes from the IFE. Others have hypothesized that subtle geographic variances in the embryonic origin of epithelial versus non-epithelial melanocytes could explain the restriction of certain driver mutations to specific melanoma subtypes (*Bastian, 2014*; *Whiteman et al., 2011*). Alternatively, since epidermal melanocytes interact with keratinocytes, whereas non-epidermal melanocytes interact with the mesodermal stromas, it was possible that direct cell contact or paracrine signaling produced by the tissue-specific microenvironment might interfere with the oncogenic signaling pathway (*Pandiani et al., 2017*). Here, we have presented evidence that the IFE microenvironment drives the loss of melanocytes expressing the GNAQ^Q209L oncogene, providing a new model for oncogene specificity.

## Melanocytes taken from the IFE and cultured in vitro can switch phenotypes

Previous studies have shown that stable transfection of *GNAQ^Q209L* induced anchorage-independent growth in soft agar of hTERT/CDK4^R24C/p53^DD mouse melanocytes in a TPA-independent manner, a feature associated with cellular transformation (*Van Raamsdonk et al., 2009*; *Wilson et al., 1989*). Our study complements these previous results to show that melanocytes directly taken from the mouse

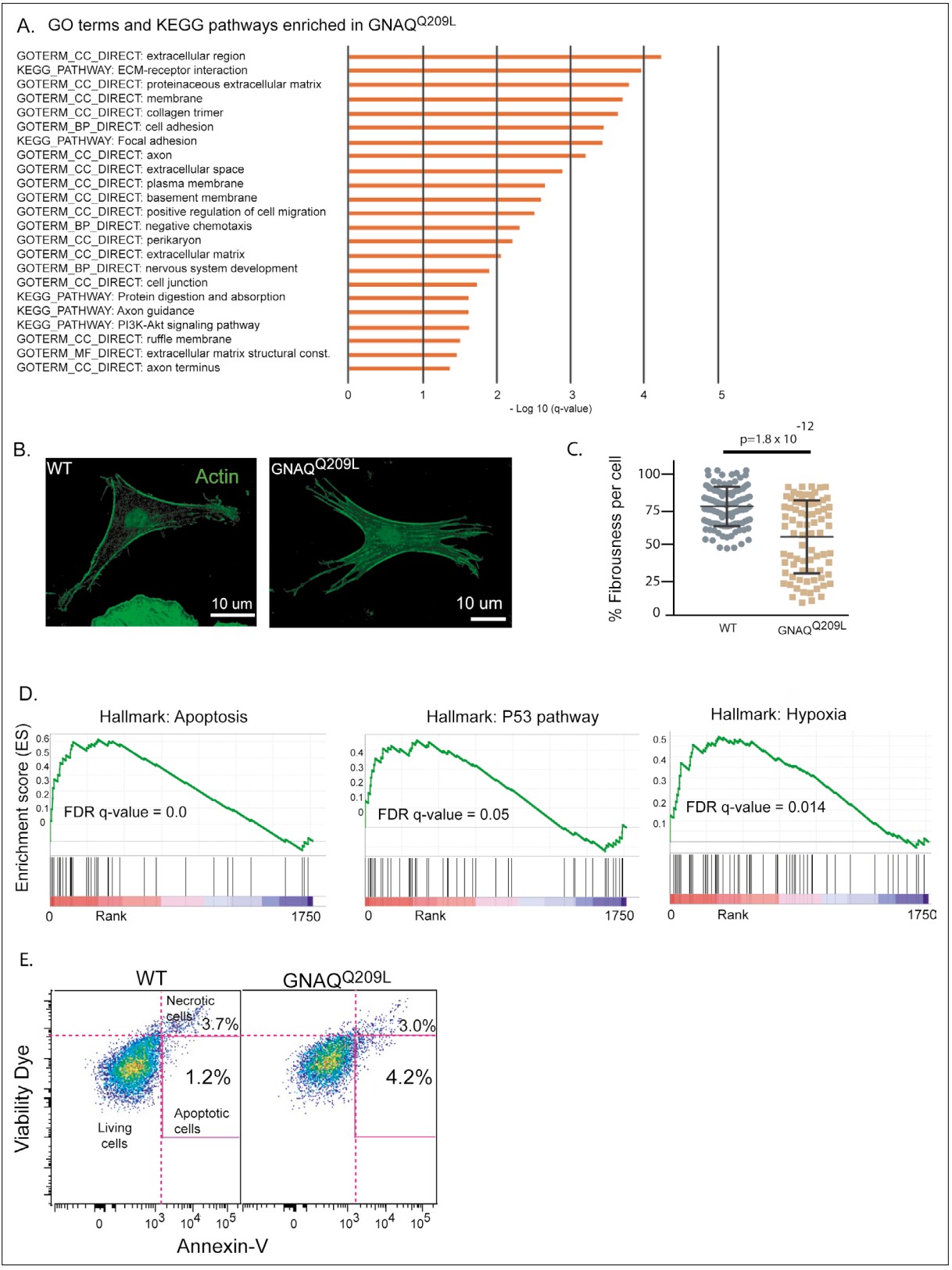

**Figure 6.** Analysis of GNAQ^Q209L expressing melanocytes in the interfollicular epidermis (IFE) reveals alterations in the actin cytoskeleton and cell death. (**A**) Significant terms identified by Gene Ontology analysis for differentially expressed (DE) genes (log$_2$ fold change [LFC] >2 or <−2) in GNAQ^Q209L melanocytes. All were enriched in the GNAQ^Q209L melanocytes (red). (**B**) Representative examples of phalloidin staining for f-actin in wildtype (WT) and GNAQ^Q209L melanocytes co-cultured with IFE. The actin is less organized in the GNAQ^Q209L cell. (**C**) Reduced fibrousness in GNAQ^Q209L cells indicates

*Figure 6 continued on next page*

*Figure 6 continued*

disorganization of the actin cytoskeleton. (Each point represents a measurement from one cell, mean ± SEM; unpaired t test). (**D**) Gene set enrichment plots for apoptosis, P53 pathway, and hypoxia hallmarks enriched in GNAQ$^{Q209L}$ melanocytes. (**E**) tdTomato+ cells identified by fluorescent activated cell sorting (FACS) from 4-week-old tail IFE and sorted for Annexin-V and viability. N = 2 WT and 2 GNAQ$^{Q209L}$ mice, melanocytes pooled. Early apoptotic cells are positive for Annexin-V and negative for the viability dye. Note, the necrotic cells are cells that had a disrupted membrane that allowed entry of the viability dye. They should not be used to assess apoptosis because Annexin-V is able to enter damaged cells and bind phosphatidylserine on the inner leaflet, creating a false positive.

IFE and grown in primary culture without any other cues survive better if they express *GNAQ$^{Q209L}$*. Interestingly, *Braf$^{V600E}$* had an even stronger effect than *GNAQ$^{Q209L}$* in this same situation, although the promoters driving the expression of these two oncogenes were not the same, so perhaps a direct comparison should not be made. In addition, both WT and GNAQ$^{Q209L}$ expressing melanocytes survived better in culture when grown on MEFs. Therefore, the attrition of GNAQ$^{Q209L}$ expressing IFE melanocytes from the mouse tail epidermis is not due to an inherent characteristic in IFE melanocytes because the phenotype is reversible by changing the microenvironment.

## The IFE microenvironment inhibits melanocytes expressing GNAQ$^{Q209L}$

It is well known that keratinocytes, which make up the vast majority of the epidermis, secrete various growth factors and cytokines in a paracrine manner that regulates growth, survival, adhesion, migration, and differentiation of melanocytes (*Wang et al., 2016*). In WT melanocytes, we found that the presence of dissociated IFE in co-cultures provided a significant boost. In contrast, co-culturing GNAQ$^{Q209L}$ expressing melanocytes with IFE reduced their survival capacity and inhibited cell division. Moreover, the effects of the IFE on melanocyte survival did not require direct cell-cell contact and could be replicated in a transwell culture system. Our studies of IFE melanocytes suggest that GNAQ$^{Q209L}$ expression causes increased cell adhesion, a disorganized actin cytoskeleton and the promotion of long dendrite extensions, which frequently break. We conclude that the microenvironment surrounding melanocytes is the main factor controlling whether GNAQ$^{Q209L}$ signaling is oncogenic or inhibits growth. IFE melanocytes have the innate capacity to be transformed by GNAQ$^{Q209L}$, but whether or not they are permitted to do so is dictated by the microenvironment in which they reside.

## A cellular stress response signature in GNAQ$^{Q209L}$ IFE melanocytes related to vitiligo

Reactive oxygen species (ROS) are highly active radicals produced during multiple cellular processes that, upon accumulation, can damage most biological macromolecules (*Bickers and Athar, 2006*). ROS overproduction in melanocytes by exogenous and endogenous stimuli has been implicated in vitiligo, a common skin depigmentation disorder caused by a loss of melanocytes from the epidermis. ROS accumulation can lead to membrane peroxidation, decreased mitochondrial membrane potential, and apoptosis of melanocytes in vitiligo epidermis (*Bellei et al., 2013*). Interestingly, our RNAseq analysis of GNAQ$^{Q209L}$ IFE melanocytes revealed changes in gene expression that suggest the melanocytes are under stress. GSEA found that GNAQ$^{Q209L}$ melanocytes are enriched in the hallmarks for 'apoptosis', 'p53 pathway', and 'hypoxia'. *Cdkn2a* (p16) and Ccnd1 (cyclin D1) were both up-regulated in expression (*Bellei et al., 2013*; *Becatti et al., 2014*; *Choi et al., 2010*). Previous studies have shown that melanocytes have an increased susceptibility to oxidative stress compared with keratinocytes or fibroblasts and that p16 plays a crucial role in regulating oxidative stress independently of Rb tumor suppressor function (*Jenkins et al., 2011*). Furthermore, oncogene activation in melanocytes generates oxidative stress and increased ROS and p16 expression has been implicated in oncogene-induced melanocyte senescence (*Leikam et al., 2008*). Thus, up-regulation of Cdkn2a may reflect a systemic oxidative stress experienced by *GNAQ$^{Q209L}$* melanocytes in an epithelial context.

In addition, among the top-ranked DE genes was *Stanniocalcin* (Stc1), a glycoprotein hormone involved in calcium/phosphate homeostasis. Significant up-regulation of *STC1* was reported in tumors under hypoxic or oxidative stress (*Yeung et al., 2005*; *Nguyen et al., 2009*; *Guo et al., 2013*; *Block et al., 2009*; *Lai et al., 2007*). Furthermore, *Stc1* expression down-regulates pro-survival Erk1/2 signaling and reduces survival of MEFs under conditions of oxidative stress (*Nguyen et al., 2009*). Using FACS, we found that there are more melanocytes in early apoptosis in the GNAQ$^{Q209L}$ tail IFE

**Figure 7.** Model for microenvironmental control of the Gα$_q$-PLCB4 signaling outcome and how this leads to GNAQ oncogene specificity in melanoma. G alpha q (GNAQ) and G alpha 11 (GNA11) are classic signaling components that transmit G protein-coupled receptor (GPCR) activation to phospholipase C-beta (PLC-B) inside the cell. Either GNAQ, GNA11, or PLCB4 is activated by a gain-of-function (GOF) hotspot mutation in the majority of uveal (ocular) and CNS melanomas, but these mutations are almost never found in melanomas arising in the epidermis of the skin. We propose that the reason for this is that paracrine signaling from the epidermis reversibly switches Gα$_{q/11}$ signaling from promoting growth to inhibiting melanocyte survival and proliferation. GNAQ$^{Q209L}$, in combination with signaling from the interfollicular epidermis (IFE), stimulates dendrite extension, leads to actin cytoskeleton disorganization, inhibits proliferation, and promotes apoptosis in melanocytes.

than in wildtype (WT). This represents a novel and intriguing potential consequence of GNAQ[Q209L] signaling induced by the IFE microenvironment.

## Cell morphology and endothelin signaling

GNAQ[Q209L] signaling produces an abnormal cell morphology with irregular contours. This has been previously described in *GNAQ[Q209L]*-transformed human melanocytes and melanocytes of a *Gnaq[Q209L]* zebrafish model (*Perez et al., 2018*). In our current study, GNAQ[Q209L] melanocytes taken from the IFE initially appeared similar to WT melanocytes when plated, but became very large and abnormally shaped when co-cultured with MEFs (*Figure 2I*). However, GNAQ[Q209L] melanocytes co-cultured with IFE were also larger and abnormally shaped (*Figure 3D and G*). Our RNAseq results from GNAQ[Q209L] melanocytes taken directly from the IFE suggest that cell adhesion, focal adhesions, extracellular matrix-receptor interactions, and/or extracellular matrix structural constituents are impacted. For example, *Collagens type IV*, *V*, and *XXVII*, *Fn1*, *Itga1*, *Itgb3*, *Amigo1*, *Mcam*, *Cadherin 6*, *Cercam*, and *Esam* were all up-regulated in the GNAQ[Q209L] melanocytes. Confocal z-projections of individuals melanocytes processed using quantitative image analysis to measure cell fibrousness showed that the actin cytoskeleton of GNAQ[Q209L] melanocytes co-cultured with IFE is disorganized. Since cell size and shape were abnormal in GNAQ[Q209L] melanocytes in both the IFE and MEF co-cultures, it is possible that this GNAQ[Q209L] phenotype is less sensitive to microenvironmental control.

Some of the changes that we observed with GNAQ[Q209L] could reflect a normal role for G protein-coupled endothelin signaling in epidermal melanocytes. *Belote and Simon, 2020*, used human keratinocyte-melanocyte co-cultures to show how endothelin and acetylcholine secreted by keratinocytes elicit local and compartmentalized $Ca^{2+}$ transients in melanocyte dendrites (*Belote and Simon, 2020*). Endothelin receptors can couple to $G\alpha_{q/11}$ subunits and a melanocyte-specific *Ednrb* (Endothelin receptor B) knockout reduced the effects of GNAQ[Q209L] expression in mice (*Jain et al., 2020*; *Urtatiz and Van Raamsdonk, 2016*). A recent GWAS/eQTL found that increased *EDNRB* expression was associated with lighter colored (blond) hair in Canadians with European ancestry (*Lona-Durazo et al., 2021*).

In *Figure 7*, we summarize our model for *GNAQ/11* oncogene specificity in melanoma. We suggest that the outcome of $G\alpha_{q/11}$ signaling is not fixed, but is integrated with other information provided from the microenvironment in which the melanocyte resides. Cues from the IFE cause $G\alpha_{q/11}$ signaling to inhibit survival and proliferation, but this effect is reversible. In the absence of these external cues, $G\alpha_{q/11}$ signaling promotes melanocyte survival. Hence, gain-of-function mutations in *GNAQ*, *GNA11*, or *PLCB4* only drive melanoma in melanocytes surrounded by mesenchymal stromas, and not by keratinocytes. Our studies revealed evidence for multiple potential mechanisms by which the IFE inhibits melanocytes expressing GNAQ[Q209L], including oxidative stress, apoptosis, inhibition of cell division, changes to cell adhesion, and cell fragmentation. The critical paracrine signal(s) remain to be determined through further biochemical and cell culture studies. It is possible that a better understanding of the ability of GNAQ[Q209L] to switch from acting as an oncogene to an inhibitor of melanocyte growth could lead to new therapies for *GNAQ* and *GNA11* mutant melanomas, which lack effective treatment options for metastatic disease.

# Materials and methods

## Key resources table

| Reagent type (species) or resource | Designation | Source or reference | Identifiers | Additional information |
|---|---|---|---|---|
| Strain background (*Mus musculus domesticus*, males and females) | C3HeB/FeJ | Jackson Laboratories | Jackson Labs: strain #000658 | |
| Gene (*Homo sapiens*) | GNAQ | GenBank | GenBank Gene ID: 2776 | |

*Continued on next page*

*Continued*

| Reagent type (species) or resource | Designation | Source or reference | Identifiers | Additional information |
|---|---|---|---|---|
| Genetic reagent (*Mus musculus domesticus* males and females) | Tg(Mitf-cre) 7114Gsb | *Alizadeh et al., 2008* | PMID:18353144 RRID:MGI:5702900 | |
| Genetic reagent (*Mus musculus domesticus* males and females) | Tg(Tyr-cre/ ERT2)13Bos/J | Jackson Laboratories | Jackson Labs: strain #012328 | |
| Genetic reagent (*Mus musculus domesticus* males and females) | Gt(ROSA) 26Sor$^{tm1(GNAQ*)Cvrk}$ | *Huang et al., 2015* | PMID:26113083 RRID:MGI:5702877 | |
| Genetic reagent (*Mus musculus domesticus* males and females) | Gt(ROSA) 26Sor$^{tm14(CAG-tdTomato)Hze}$ | Jackson Laboratories | Jackson Labs: strain #007914 | |
| Genetic reagent (*Mus musculus domesticus* males and females) | Gt(Rosa) 26Sor$^{tm1Sor}$/J | Jackson Laboratories | Jackson Labs: strain #003474 | |
| Genetic reagent (*Mus musculus domesticus* males and females) | *Braf$^{tm1Mmcm}$* | Jackson Laboratories | Jackson Labs: strain #017837 | |
| Commercial assay or kit | Annexin-V-FLUOS Staining Kit | Millipore Sigma | MilliporeSigma SKU:11858777001 | |
| Commercial assay or kit | DNeasy Blood and Tissue kit | Qiagen | Qiagen ID: 69504 | |
| Commercial assay or kit | HotStar Taq DNA polymerase | Qiagen | Qiagen ID: 203207 | |
| Chemical compound, drug | Tamoxifen | Millipore Sigma | MilliporeSigma catalog:T5648 | |
| Chemical compound, drug | 4-Hydroxyt amoxifen | Millipore Sigma | Millipore Sigma catalog:H6278 | |
| Software, algorithm | Custom Matlab scripts | *Haage et al., 2018* | PMID:30485809 | https://github.com/Tanentzapf-Lab/ActinOrganization_CellMorphology_Haage |
| Other | eBioscience Fixable Viability Dye eFluor 450 | ThermoFisher | ThermoFisher catalog : 501128817 | |
| Other | RiboLock RNase inhibitor | ThermoFisher | ThermoFisher catalog :EO0381 | |
| Other | Fibronectin | Millipore Sigma | MilliporeSigma catalog:F0895 | Liquid, 0.1% solution |

## Mice

Animal research was conducted under the approval of the UBC Animal Care Committee (Protocols A18-0080 and A19-0148, CDVR). DNA from ear notches was isolated using DNeasy columns (Qiagen) and amplified using PCR. *Mitf-cre* (*Tg(Mitf-cre)7114Gsb*), *Tyr-creERT²* (*Tg(Tyr-cre/ ERT2)13Bos/J*), *Rosa26-LoxP-Stop-LoxP-GNAQ$^{Q209L}$* (Gt(ROSA)26Sor$^{tm1(GNAQ*)Cvrk}$), *Rosa26-LoxP-Stop-LoxP-tdTomato* (Gt(ROSA)26Sor$^{tm14(CAG-tdTomato)Hze}$), *Rosa26-LoxP-Stop-LoxP-LacZ* (Gt(Rosa)26Sor$^{tm1Sor}$/J), and *Braf$^{CA}$* (*Braf$^{tm1Mmcm}$*) mice were genotyped as previously described (*Huang et al., 2015*; *Alizadeh et al., 2008*; *Bosenberg et al., 2006*; *Soriano, 1999*; *Dankort et al., 2007*; *Madisen et al., 2010*). All strains were backcrossed to the C3HeB/FeJ genetic background for at least three generations.

## LacZ staining

We crossed $R26\text{-}fs\text{-}GNAQ^{Q209L}/+$; +/+ mice to $R26\text{-}fs\text{-}LacZ/+$; $Tyr\text{-}creERT^2/+$ mice and genotyped the resulting progeny. At 4 weeks of age, two groups of mice were TM-treated for 5 consecutive days: $R26\text{-}fs\text{-}GNAQ^{Q209L}/R26\text{-}fs\text{-}LacZ$; $Tyr\text{-}creER/+$ (GNAQ-LacZ) mice and $+/R26\text{-}fs\text{-}LacZ$ mice; $Tyr\text{-}creER/+$ (WT-LacZ) mice. TM treatment consisted of one daily intraperitoneal injection of 1 mg TM (Sigma T5648) alongside a topical treatment for the tail skin (tails were dipped for 5 s in 25 mg/ml 4-hydroxytamoxifen [Sigma H6278] in DMSO). Mice were harvested either 1 or 8 weeks after TM treatment. At each experimental endpoint, a piece of tail skin of 1.5 cm in length was incubated in X-gal solution for 48 hr as previously described (*Tharmarajah et al., 2018*; *Tharmarajah et al., 2012*). The tail dermis and epidermis were split using sodium bromide and fixed in 4% paraformaldehyde. The number of LacZ-positive cells was counted in three rows of epidermal scales per sample.

## Histochemistry

For H&E staining, mouse tail skin samples were fixed in 10% buffered formalin overnight at room temperature with gentle shaking, dehydrated, cleared, and embedded in paraffin; 5 µm sections were taken for H&E staining using standard methods and imaged using a DMI 6000B microscope (Leica). To examine tdTomato expression in sections, samples were fixed in 10% buffered formalin overnight at 4°C, taken through a sucrose gradient, embedded in OCT, and sectioned at 10 µm. After washing in PBS for 30 min, sections were counter-stained with DAPI and imaged using a Histotech III slide scanner.

## Isolation of interfollicular melanocytes from mouse tail skin

We crossed $R26\text{-}fs\text{-}GNAQ^{Q209L}/+$ mice to $+/R26\text{-}fs\text{-}tdTomato$; $Mitf\text{-}cre/+$ mice and identified tdTomato expressing progeny by genotyping. At 4 weeks of age, the tails were waxed to remove hair follicles. The next day, tail skin from $GNAQ^{Q209L}$ and WT tdTomato-positive mice was harvested, and the IFE was split from the dermis by gentle dispase treatment. The IFE was then incubated with trypsin into single cells, using forceps to scrape cells from the scales to accelerate the process. (A detailed description of this method is under review at *Bio Protocol,* Pop .) Cells were centrifuged and the pellet was resuspended in suspension buffer (HBSS +10% FBS+ EDTA 0.1 mM EDTA) prior to FACS.

## FACS

Dissociated cells were first analyzed for viability based on forward and side scatter plots and using eBioscience Fixable Viability Dye (ThermoFisher 501128817). Next, filters for forward scatter (height versus width) and side scatter (height versus width) were used to select single cells. Last, viability dye fluorescence (y-axis) was plotted against tdTomato (x-axis) and gates were set to capture only viable tdTomato-positive cells. Cells from mice lacking *Mitf-cre* ($+/R26\text{-}fs\text{-}tdTomato$; +/+) were used as negative controls to define the threshold for tdTomato-positive cells. Sorted cells were collected in 15 µl of HBSS + FBS 10% + EDTA 0.1 mM solution and 1 µl of RiboLock RNase inhibitor (Thermo Scientific, EO0381). For Annexin-V staining, we used the Annexin-V Fluos staining kit according to the manufacturer's directions (Sigma Aldrich, 11858777001).

## Primary cell culture

Cells were plated into 96-well plates previously coated with 0.1 mg/ml fibronectin in Dulbecco's modified Eagle's medium supplemented with 10% fetal bovine serum, 1% penicillin-streptomycin (15140122 Thermo-Fisher), 2 mM L-glutamine, and 1 mM sodium pyruvate. For cultures of sorted melanocytes only, cells were plated at 6000–8000 cells per well. For co-culture with MEFs, 6000–8000 FACS sorted melanocytes were plated into 96-well plates previously seeded with MEFs that had formed a confluent monolayer. For direct co-culture experiments with IFE, the IFE was dissociated and plated at 100,000 cells per well (i.e. with no FACS for the melanocytes first). For transwell co-culture with IFE, 6000–8000 FACS sorted melanocytes were plated in the well suspended above 100,000 dissociated IFE cells plated below. All cultures were incubated at 37°C in 5% $CO_2$. Media was changed every third day by exchanging one-third of the existing volume. Images were taken at 5×, 10×, and 20× magnification. tdTomato-positive cells in the 5× field of view were counted as live melanocytes.

## Live-cell imaging

Sorted tdTomato-positive melanocytes or IFE cells containing tdTomato-positive melanocytes were plated on 0.1 mg/ml fibronectin-coated coverslips within a 35 mm cell culture dish. Cells were

incubated overnight before imaging in a live-cell imaging culture chamber. Cells were imaged at 37°C (5% $CO_2$) every 5 min for 20 hr at 10× magnification. Cell movements were determined using the MTrackJ software on ImageJ and analysis was performed using Chemotaxis and Migration Tool (Ibidi).

## Automated image analysis

Custom Matlab scripts were utilized to analyze cell morphology and actin fiber orientation in fixed cells as described in *Haage et al., 2018*, and posted to GitHub (https://github.com/Tanentzapf-Lab/ActinOrganization_CellMorphology_Haage; copy archived at swh:1:rev:9cb24da5652e47684e73fe-36b37e88b1956f0a6b). Cell contours were obtained automatically by processing confocal z-projections of cells stained for F-actin (Phalloidin) in Matlab. Images were first blurred with a Gaussian filter, and then an edge detection algorithm was applied to identify cell borders. The resultant binary images were refined through successive dilations and erosions to yield the final cell contour. These contours were used to measure cell area, and circularity ($4\pi*area/perimeter^2$), protrusions. The number and length of protrusions were quantified automatically in Matlab. First, cell contours were identified as outlined above. Next, we obtained the contour coordinates of the convex hull of the binary image representing the cell area. At each point along the cell contour, we computed the minimum distance between the convex hull and the actual cell contour. Based on these distances, minima corresponding to protrusions could be identified. To be counted as protrusions, minima had to be at least 10 pixels apart along the contour and of height greater than five pixels. Based on the coordinates of adjacent peaks, the width, height, and aspect ratio of protrusions could be computed.

## Actin fiber organization

First, we identified cell contours as described above. Next, the cell was subdivided into 32 × 32 pixel windows overlapping by 50%. We then computed the two-dimensional Fourier transform of each window. If a window contains no fibers, the Fourier transform will be a central, diffuse point of bright pixels. However, if a window contains aligned fibers, the Fourier transform will consist of an elongated accumulation of bright pixels at a 90-degree angle to the original fibers. Based on the aspect ratio and orientation of the Fourier transform, we determined fibrousness and fiber orientation in a given window. The data for individual windows could then be compared across the entire cell to estimate the cell fibrousness, defined here as the percentage of cell area (% of windows) with aspect ratio greater than a cut-off value.

## RNAseq data and bioinformatics analysis

Total RNA from sorted cells was extracted using Trizol (Life Technologies) following the manufacturer's protocol. For each library preparation, sorted melanocytes from two mice of the same litter were pooled to increase the number of cells for RNA extraction. We generated three WT libraries and three GNAQ libraries from a total of 12 mice. Sample quality control was performed using the Agilent 2100 Bioanalyzer. Qualifying samples (RNA Integrity Number ≥9) were then prepped following the standard protocol for the NEB next Ultra ii Stranded mRNA (New England Biolabs). Sequencing was performed on the Illumina NextSeq 500 with Paired End 42 bp × 42 bp reads. Sequencing data was demultiplexed using Illumina's bcl2fastq2. Demultiplexed read sequences were then aligned to the *Mus musculus* mm10 reference sequence using STAR aligner (*Dobin et al., 2013*). The fastq sequences have been deposited at the Sequencing Read Archive (SRA) of the NCBI under BioProjectID PRJNA736153.

Assembly and differential expression analysis were performed using Cufflinks (*Trapnell et al., 2012*) through bioinformatics apps available on Illumina Sequence Hub. GO and KEGG pathways analysis was performed using DAVID Bioinformatics Resources 6.8 (*Huang et al., 2009a*; *Huang et al., 2009b*). GSEA was performed using the JAVA GSEA 2.0 program (*Subramanian et al., 2005*). The gene sets used for analysis were the Broad Molecular Signatures Database gene sets H (Hallmark gene sets).

## Statistical analysis

### Sample-size estimation

Target sample sizes were not explicitly computed during the study design phase because the standard deviations between WT and GNAQ[Q209L] epidermal melanocytes were unknown. However, in our past experience of studying skin pigmentation phenotypes on inbred genetic backgrounds, four mice of

each genotype is usually sufficient to detect statistically significant differences. For primary cell culture experiments, we planned to perform each experiment in three complete and independent runs, from FACS collection to cell culture. For RNAseq, we planned to generate three WT libraries and three GNAQ[Q209L] libraries, based on previously published studies.

### Replicates

All replicates were biological replicates, that is, each sample or independently derived cell culture was established from a different mouse. In some experiments, one biological replicate contained melanocytes collected from two mice of the same age and genotype, which were pooled together to form one sample. When pooling occurred, it was done at the FACS step. The number of replicates (number of mice and number of melanocytes derived from the mice) can be found in *Supplementary file 2d*, organized by figure number.

### Statistical reporting

The statistical tests that were used in each figure are described and shown in *Supplementary files 2d and 3*. Raw data was plotted in graphs. Except for the survival graphs, the mean for each group and the standard error of the mean is shown. Exact p values, t values, degrees of freedom, and number of replicates are detailed in *Supplementary file 2d*. Statistical analysis was performed using GraphPad Prism software. For RNAseq, the magnitude and significance of differential gene expression and multiple test corrections were determined using the Cufflinks suite via the Illumina Sequence Hub. GO and KEGG pathways analysis was performed using DAVID Bioinformatics Resources 6. GSEA was performed using the JAVA GSEA 2.0 program.

### Group allocation

Samples were allocated into experimental groups based on genotype (WT or GNAQ[Q209L]) or culture condition. WT and GNAQ[Q209L] mice/melanocytes that were compared to each other were produced from the same set of breeder parents. Masking was used whenever possible (melanocytes/skin samples with an obvious difference in pigmentation are not maskable).

## Acknowledgements

We thank Dr Gregory Barsh for contributing the *Mitf-cre* mouse line. Research was supported by grants from the Canadian Institutes of Health Research (grant MOP-79511 to CDVR and grant PJT-168868 to GT).

## Additional information

### Funding

| Funder | Grant reference number | Author |
|---|---|---|
| Canadian Institutes of Health Research | MOP-79511 | Catherine D Van Raamsdonk |
| Canadian Institutes of Health Research | PJT-168868 | Guy Tanentzapf |

The funders had no role in study design, data collection and interpretation, or the decision to submit the work for publication.

### Author contributions

Oscar Urtatiz, Conceptualization, Investigation, Methodology, Visualization, Writing – original draft, Writing – review and editing; Amanda Haage, Conceptualization, Investigation, Methodology, Writing – review and editing; Guy Tanentzapf, Conceptualization, Funding acquisition, Resources, Software, Supervision, Writing – review and editing; Catherine D Van Raamsdonk, Conceptualization, Funding acquisition, Investigation, Project administration, Resources, Supervision, Visualization, Writing – original draft, Writing – review and editing

Author ORCIDs
Amanda Haage ⓘ http://orcid.org/0000-0001-6305-440X
Guy Tanentzapf ⓘ http://orcid.org/0000-0002-2443-233X
Catherine D Van Raamsdonk ⓘ http://orcid.org/0000-0002-4309-3513

## Ethics

Animal research was conducted under the approval of the University of British Columbia Animal Care Committee (Protocols A18-0080 and A19-0148, C.D.V.R.).

## Decision letter and Author response

Decision letter https://doi.org/10.7554/eLife.71825.sa1
Author response https://doi.org/10.7554/eLife.71825.sa2

## Additional files

### Supplementary files

• Supplementary file 1. Excel file providing more details of the differentially expressed genes in GNAQ$^{Q209L}$ melanocytes sorted from the IFE.
(a) Genes with FPKM >0.1 in wildtype (WT) and/or GNAQ$^{Q209L}$ melanocytes. (b) Differentially expressed genes down-regulated in GNAQ$^{Q209L}$ interfollicular epidermis (IFE) melanocytes, sorted by Z_score. (c) Differentially expressed genes up-regulated in GNAQ$^{Q209L}$ IFE melanocytes, sorted by Z_score. (d) Genes supporting pathway analysis terms related to cell adhesion, focal adhesion, and the extracellular matrix. (e) Genes supporting pathway analysis terms related to axon guidance, nervous system development, and axon cellular component.

• Supplementary file 2. Word file providing more details on gene expression in mouse tail IFE melanocytes, frequency of oncogenic mutations in *GNAQ* and *GNA11* in human cutaneous melanomas, and statistical tests used in the studies.
(a) Top 20 most highly expressed genes in mouse wildtype (WT) interfollicular epidermis (IFE) melanocytes. (b) Identification of *GNAQ* hotspot mutations among human malignant melanomas potentially arising in the epidermis. (c) Identification of *GNA11* hotspot mutations among human malignant melanomas potentially arising in the epidermis. (d) Information on statistical tests in *Figures 1–6*.

• Supplementary file 3. Prism file with source data used to create graphs and calculate statistics in *Figures 1–6*.

• Transparent reporting form

### Data availability

Sequencing data has been deposited at the Sequencing Read Archive (SRA) of the NCBI under BioProjectID PRJNA736153. The custom MATLAB scripts have been deposited to GitHub at https://github.com/Tanentzapf-Lab/ActinOrganization_CellMorphology_Haage (copy archived at swh:1:rev:9cb-24da5652e47684e73fe36b37e88b1956f0a6b). All other data generated or analysed during this study are included in the manuscript and supporting files.

The following dataset was generated:

| Author(s) | Year | Dataset title | Dataset URL | Database and Identifier |
|---|---|---|---|---|
| Urtatiz O, Haage A, Tanentzapf G, Van Raamsdonk CD | 2022 | Effects of oncogenic GNAQ on epidermal melanocytes | https://www.ncbi.nlm.nih.gov/sra/PRJNA736153 | NCBI Sequence Read Archive, PRJNA736153 |

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
