## [Editor Report]

Urtatiz and colleagues propose that gain-of-function mutations affecting the G-α-q signaling pathway are not tolerated in melanocytes residing in the interfollicular epidermis because of paracrine signals from neighboring keratinocytes. This is an interesting and important hypothesis that would explain a mystery to the melanoma field.

---

## [Decision Letter]

**Decision letter after peer review:**

Thank you for submitting your article "Crosstalk with keratinocytes causes GNAQ oncogene specificity in melanoma" for consideration by *eLife*. Your article has been reviewed by 2 peer reviewers, and the evaluation has been overseen by a Reviewing Editor and Richard White as the Senior Editor. The reviewers have opted to remain anonymous.

Essential revisions:

1) The bioinformatic analysis regarding PLCB4 is flawed. This analysis requires a more sophisticated analysis or should be removed from the manuscript. As currently presented, this detracts from the conclusions rather than supporting the work.

2) The Semaphorin mechanism is interesting but highly speculative.

3) The authors should address some inconsistencies around the data presented in Figure 3A and 4D. It seems like the WT-melanocytes in co-culture with IFE are dividing quite rapidly, according to figure 4D, yet their numbers decline over time according to figure 3A.

4) The fluorescence images would be better with higher quality images.

*Reviewer #2 (Recommendations for the authors):*

The authors have utilized appropriate models and multiple experimental methods to support their findings.

The only weakness noted in the lengthy discussion of the TCGA-SKCM data set analysis and comparison of the frequency of expression of mutations in various genes. This part needs to be summarized, since it distracts from the major take-home message of the manuscript and the strong data demonstrating how the microenvironment can influence the oncogenic impact of GNAQ mutations.

[Editors' note: further revisions were suggested prior to acceptance, as described below.]

Thank you for resubmitting your work entitled "Crosstalk with keratinocytes causes GNAQ oncogene specificity in melanoma" for further consideration by *eLife*. Your revised article has been evaluated by Richard White (Senior Editor) and a Reviewing Editor.

The manuscript has been improved but there is one remaining issue that need to be addressed, as outlined below:

The reviewers and editors are not convinced that the mutational analysis of PLCB4 are robust and follow currently accepted guidelines for identifying somatic alterations in tumors with high numbers of somatic alterations, such as melanoma. The cited literature is not mainstream. Since this point is not necessary to support the conclusions presented, we ask that you remove this analysis from the manuscript.

*Reviewer #1 (Recommendations for the authors):*

I compliment the authors for clarifying the issues pertaining to their experimental work. I also think the addition to the manuscript of the MAPK and Hippo pathway gene expression analyses was very interesting. Questions have been raised by other laboratories as to the relevance of the Hippo arm, and these results will be informative to that debate.

I am still not convinced that PLCB4 mutations are under selection, and I would strongly encourage the authors to abandon the claim in this manuscript, which I do not believe is necessary for publication in *eLife*. Aside from this claim, I really enjoyed this manuscript, and in the long run, I think the authors would regret this position without more supporting data. My colleagues and I have spent a great deal of time thinking about the best ways to identify genes under selection in melanoma (see https://onlinelibrary.wiley.com/doi/full/10.1111/pcmr.12012), and I encourage the authors to reach out if they have any questions.

In this revision, the main reason why the authors were able to claim the dN/dS ratio of mutations affecting PLCB4 was statistically significant is because they did not include mutations in tumors with both nonsynonymous and synonymous mutations. I have never seen anyone apply this filter before. Given that nonsynonymous mutations are more common across the genome than synonymous mutations, removal of collision events will disproportionately impact the proportion of synonymous mutations, thus skewing the ensuing ratio of nonsynonymous to synonymous mutations. It is not statistically defensible to apply a filter to the observed mutational data in a single gene of interest and then to compare it to an expected background mutation rate where the same correction was not applied. The authors attempt to justify this decision by speculating that synonymous mutations likely occurred after the nonsynonymous mutations – this does not seem to occur at a meaningful degree in well established tumor suppressor genes and is far too speculative to use as justification.

The authors also point out that there were some splice-site mutations and nonsense mutations, but again, the ratio of these mutations to missense mutations and synonymous mutations was not particularly impressive. Truncating mutations (nonsense, frameshift, and splice-site) are extremely common in well-established tumor suppressor genes, often exceeding 50% of the total mutation count. If anything, the low proportion of these mutations is further evidence that there is little selection to accumulate bona fide loss-of-function mutations in this gene, but rather the locus collects a high background of passenger mutations.

The authors cite a brief mention in the supplement of a manuscript, Chen et al. (Mol Biol Evol 2015) as support that PLCB4 has a statistically significant number of splice-site mutations. The methodologies in that manuscript are not robust, nor are their candidates convincing, likely explaining why this obscure manuscript has gained no traction in the genomics community.

The authors also cite a manuscript by Wei et al. (2012), who claimed PLCB4 mutations were statistically significant. This was one of the earliest exome sequencing studies in melanoma, and the field generally agrees that their approaches to identifying significantly mutated genes was flawed. More modern cancer gene discovery studies have not identified evidence of selection in PLCB4.

The authors additionally cite a mention from a manuscript by Hayward et al. (2017), who reported gene fusions affecting PLCB4. In that manuscript, Hayward and colleagues list 30,000 potential gene fusions, and a handful affect PLCB4. It is unclear if these fusions are real because they were not validated. While Hayward and colleagues specifically mention PLCB4 in their results, they imply that the fusions might be gain of function, whereas in this manuscript, the authors are attempting to claim melanomas select for loss of function somatic alterations.

The bottom line is that the mutational patterns in PLCB4 look unlike any well-established tumor suppressor gene in melanoma. This reviewer recognizes that the authors came across this candidate as a result of hypothesis driven research, but it would be more compelling if PLCB4 were a borderline candidate, overlooked in high-quality, genome-wide studies that were agnostic to biological knowledge, but this does not seem to be the case. In the melanoma cancer genome atlas project, the ratio of nonsynonmous to synonymous mutations affecting PLCB4 was 2.8, which is almost identical to the genome-wide ratio, thereby making the dN/dS ratio for this gene approximately 1. In the melanoma genome atlas study, PLCB4 ranked ~12,000 among significantly mutated genes. In a more recent study that includes over 1,000 melanomas (Alkallas, Nature Cancer, 2020), PLCB4 ranked 9,000 among significantly mutated genes.

The functional data pertaining to GNAQ in the first three quarters of the paper was compelling, but the authors overreached, unnecessarily, in pushing a mechanism by which there is selection to lose a downstream component of this signaling pathway in melanoma. The authors show that turning on the Gq signaling pathway, via a powerful gain-of-function mutation in GNAQ, does not permit melanocyte survival in the epidermis of mice. Basal levels of Gq signaling, in the absence of Gq mutations, is low, and there is no reason to assume that tumors would need to select for mutations to turn this pathway off by accumulating mutations downstream of Gq. As it stands, this is a major claim, weakly supported entirely by in silico data.

*Reviewer #2 (Recommendations for the authors):*

In this revised manuscript, the authors have appropriately responded to the concerns of the prior reviewers. As it stands, the manuscript is recommended for publication.

---

## [Author Response]

Essential revisions:1) The bioinformatic analysis regarding PLCB4 is flawed. This analysis requires a more sophisticated analysis or should be removed from the manuscript. As currently presented, this detracts from the conclusions rather than supporting the work.

We understand the concerns of the reviewer. To explain, we were led to the question of whether the frequent non-synonymous mutations in PLCB4 could play a role in cutaneous melanoma from a logical deduction based on biological insight, which we found quite convincing:

1) We had found that oncogenic GNAQ inhibits melanocyte growth/survival when the cells are located in the epidermis.

2) PLC-Β is the immediate downstream effector of Gq and therefore is very likely to play a role in this inhibition.

3) PLC-B4 was already identified as a melanoma oncogene in uveal and leptomeningeal melanomas, but strikingly, only in cases where GNAQ and GNA11 were not activated by gain of function mutations.

We agree that more could have been done to describe the synonymous mutations and examine the ratio of non-synonymous to synonymous mutations. Using the approach described in Van den Eynden and Larsson, which takes into account the mutational signature of melanoma, we found that the synonymous mutations were significantly less frequent than expected based on the expected ratio (p = 7.35 x 10^-4^; one-tailed binomial test). This indicates that there is positive selection for the non-synonymous mutations in PLCB4 in cutaneous melanoma.

In addition, not included in the above analysis were 13 more mutations having to do with splicing: 7 splice region, 3 splice donor site and 3 splice acceptor site mutations in the melanomas. While reviewing the literature, we noticed that PLCB4 was already identified in Chen et al as a gene subject to positive selection for splice-site mutations (p=7.86Ex10^-4^; q=5.77Ex10^-2^) in a genome wide analysis of melanoma.

As you know, due to a number of factors related to the process of mutagenesis, gene size and accessibility, different genes have different rates of mutation. PLCB4 might be a gene that is more likely to be mutated. It is the 99^th^ most frequently mutated gene in the SKCM dataset. PLCB4 has more synonymous mutations (34 affected cases in total) than NF1 (14 cases), TP53 (0 cases) or PTEN (0 cases). Looking at the problem from our perspective, it seems that biological insight and hypothesis driven inquiry could help determine whether some of the genes with a higher inherent rate of mutation are playing a role in melanoma.

2) The Semaphorin mechanism is interesting but highly speculative.

We have reduced the emphasis on semaphorin signaling throughout the manuscript.

3) The authors should address some inconsistencies around the data presented in Figure 3A and 4D. It seems like the WT-melanocytes in co-culture with IFE are dividing quite rapidly, according to figure 4D, yet their numbers decline over time according to figure 3A.

This is a good point. We think that the difference is due to the culture conditions in the survival curve experiments versus the time lapse imaging experiments. In the experiments in Figure 3A (survival curve with IFE), there was less media relative to cells than there was in the experiments in Figure 4D (time lapse cell imaging of cells with IFE). This was because cells in Figure 3A were cultured in a 96 well plate format, while the cells in Figure 4D were cultured in 35 mm dishes on coverslips (for microscopy). However, under both set-ups, WT melanocytes did better than GNAQ melanocytes. The different set-ups are interesting in that they showed that both proliferation and survival are impacted by GNAQ. We have added a paragraph about this in the Results section.

4) The fluorescence images would be better with higher quality images.

Better images have been provided for Figure 1F, in which the tomato signal was over-exposed.

The authors have utilized appropriate models and multiple experimental methods to support their findings.The only weakness noted in the lengthy discussion of the TCGA-SKCM data set analysis and comparison of the frequency of expression of mutations in various genes. This part needs to be summarized, since it distracts from the major take-home message of the manuscript and the strong data demonstrating how the microenvironment can influence the oncogenic impact of GNAQ mutations.

Thank you for your analysis. We appreciate your comments. We have reduced our discussion of the findings in Figure 8A into a more concise summary and tried to better integrate our conclusions from this part of the study with the rest of the paper.

[Editors' note: further revisions were suggested prior to acceptance, as described above, which related to removing the PLCB4 analysis from the manuscript]

Thank you to the reviewers and the editors for reviewing our paper.

We have removed the data and discussion that involved PLCB4 and cutaneous melanoma (TCGA-SKCM data set).